# Score-Based Generative Models Detect Manifolds

**Jakiw Pidstrigach**
Institut für Mathematik
Universität Potsdam
Karl-Liebknecht-Str. 24/25
14476 Potsdam
pidstrigach@uni-potsdam.de

## Abstract

Score-based generative models (SGMs) need to approximate the scores $\nabla \log p_t$ of the intermediate distributions as well as the final distribution $p_T$ of the forward process. The theoretical underpinnings of the effects of these approximations are still lacking. We find precise conditions under which SGMs are able to produce samples from an underlying (low-dimensional) *data manifold* $\mathcal{M}$. This assures us that SGMs are able to generate the "right kind of samples". For example, taking $\mathcal{M}$ to be the subset of images of faces, we find conditions under which the SGM robustly produces an image of a face, even though the relative frequencies of these images might not accurately represent the true data generating distribution. Moreover, this analysis is a first step towards understanding the generalization properties of SGMs: Taking $\mathcal{M}$ to be the set of all training samples, our results provide a precise description of when the SGM memorizes its training data.

## 1  Introduction

Score-based generative models, also called diffusion models ([Sohl-Dickstein et al., 2015, Song and Ermon, 2019, Song et al., 2021b, Vahdat et al., 2021]) and the related models ([Bordes et al., 2017, Ho et al., 2020, Kingma et al., 2021]) have shown great empirical success in many areas, such as image generation ([Jolicoeur-Martineau et al., 2021, Nichol and Dhariwal, 2021, Dhariwal and Nichol, 2021, Ho et al., 2022]), audio generation ([Chen et al., 2021, Kong et al., 2021, Jeong et al., 2021, Popov et al., 2021]) as well as in other applications ([Batzolis et al., 2021, De Bortoli et al., 2021, Zhou et al., 2021, Cai et al., 2020, Luo and Hu, 2021, Meng et al., 2021, Saharia et al., 2021, Li et al., 2022, Sasaki et al., 2021]). Recently some progress has been made to bridge the gap between the different approaches ([Song et al., 2021b, Huang et al., 2021]) through the framework of SDEs and reverse SDEs.

In generative modelling one is given samples $\{x^i\}_{i=1}^N$ from a measure $\mu_{\text{data}}$. The task is to learn a measure $\mu_{\text{sample}}$ which approximates $\mu_{\text{data}}$. The performance of a generative model can then be measured by the distance from $\mu_{\text{sample}}$ to $\mu_{\text{data}}$. In practice however, the true data generating distribution $\mu_{\text{data}}$ is unknown. All that is known are the samples $\{x_i\}_{i=1}^n$, which can be used to define the empirical measure $\hat{\mu}_{\text{data}}$,

$$\hat{\mu}_{\text{data}} := \text{Unif}\{x^1, x^2, \ldots, x^n\}.$$

Any sample from the empirical measure $\hat{\mu}_{\text{data}}$ will be equal to a training example. Hence, while $\mu_{\text{sample}}$ being close to $\mu_{\text{data}}$ is the final goal, $\mu_{\text{sample}}$ being close to $\hat{\mu}_{\text{data}}$ implies that the generative model has memorized the training data. To summarize, a good generative model will output a measure $\mu_{\text{sample}}$ which is as close to $\mu_{\text{data}}$ as possible, while keeping some distance from $\hat{\mu}_{\text{data}}$, even though it only knows $\mu_{\text{data}}$ through $\hat{\mu}_{\text{data}}$.

36th Conference on Neural Information Processing Systems (NeurIPS 2022).

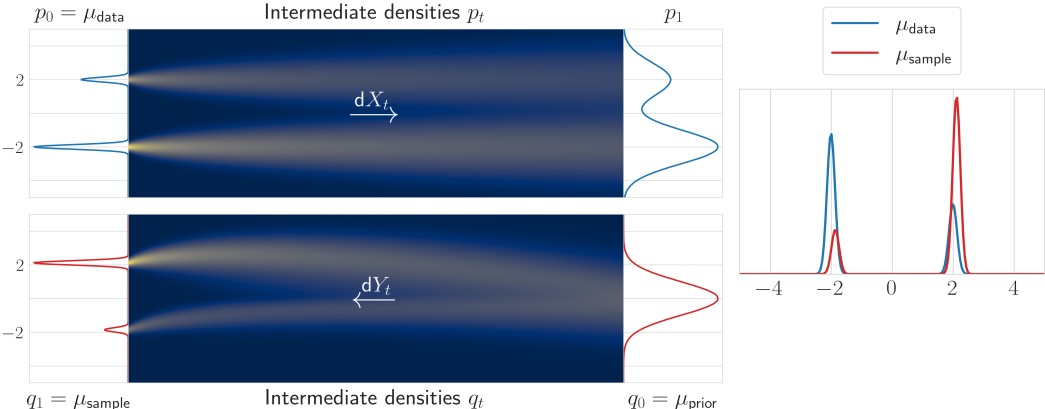

Figure 1: *Top left*: The leftmost plot shows the true data distribution $\mu_{\text{data}}$ which is a Gaussian mixture. The heat maps show the intermediate densities $p_t$ of $X_t$, followed by line plots of $p_t$ for $t = 1$. *Bottom left*: The rightmost plot shows $\mu_{\text{prior}}$, which is a standard Gaussian and differs from $p_1$. We start the reverse SDE (2) in $\mu_{\text{prior}}$. But instead of using the real score, we introduce an approximation error and use $s(x, t) = \nabla \log p_{1-t}(x) + 3$ with a constant error of 3. Again, the heat maps show how the densities $q_t$ of $Y_t$ evolve backwards in time. The leftmost plot shows the resulting distribution $q_1$ which is used as sample distribution, $\mu_{\text{sample}} = q_1$. *Right:* The densities $\mu_{\text{data}}$ and $\mu_{\text{sample}}$ are shown for direct comparison. We see that the approximation errors in $\mu_{\text{prior}}$ and the drift lead to an incorrect sample distribution $\mu_{\text{sample}} \neq \mu_{\text{data}}$. Nevertheless, $\mu_{\text{sample}}$ is supported in the same area as $\mu_{\text{data}}$. For details on the numerical implementation see Appendix B.

Given a target measure $\pi_0$, a score-based generative model (SGM) employs two stochastic differential equations (SDEs). The first one is called the *forward SDE*

$$\begin{aligned} \mathrm{d}X_t &= \beta(X_t)\mathrm{d}t + \sigma\mathrm{d}W_t, \\ X_0 &\sim \pi_0. \end{aligned} \tag{1}$$

The marginals of $X_t$ are denoted by $\pi_t$. The forward SDE is run until some terminal time $T$. Furthermore, the *reverse SDE* is defined by

$$\begin{aligned} \mathrm{d}Y_t &= -\beta(Y_t)\mathrm{d}t + \sigma\sigma^T\nabla \log p_{T-t}(Y_t)\mathrm{d}t + \sigma\mathrm{d}B_t, \\ Y_0 &\sim q_0. \end{aligned} \tag{2}$$

We refer to the marginals of $Y_t$ as $q_t$. The samples are generated from the final distribution $q_T$, i.e. $\mu_{\text{sample}} := q_T$. The reverse SDE has the property that if $q_0$ is chosen to be equal to $\pi_T$, then $q_t = \pi_{T-t}$. In particular, this implies that $q_T = \pi_0$. Therefore, if we have samples from $\pi_T$, we can run the reverse SDE on them to create new samples from $\pi_0$.

In the following we will denote by $p_t$ the marginals of the forward SDE when started in the true data generating distribution $\pi_0 = \mu_{\text{data}}$. We will denote by $\hat{p}_t$ the marginals of the forward SDE when started in $\pi_0 = \hat{\mu}_{\text{data}}$. Optimally, we would like to run the algorithm using $\pi_0 = \mu_{\text{data}}$, i.e. with marginals $\pi_t = p_t$. This is however not possible, since $\mu_{\text{data}}$ itself is unknown.

To circumvent the problem of not knowing $p_T$, the forward SDE is chosen such that it forgets its initial condition $p_0$. At time $T$, the marginal $p_T$ is then well approximated by a proxy distribution $\mu_{\text{prior}} \approx p_T$, independently of $p_0$. Additionally, the marginals $p_t$ and therefore the scores $\nabla \log p_t$ cannot be evaluated for $p_0 = \mu_{\text{data}}$. Therefore, the scores are replaced by a neural network $s_\theta(x, t)$, which is trained via score-matching techniques [Vincent, 2011, Hyvärinen and Dayan, 2005].

As a result, SGMs make two approximations. The first one is in approximating $p_T$ by $\mu_{\text{prior}}$. The second one is the approximation of $\nabla \log p_t$ by the neural net $s_\theta(x, t)$. We illustrate this in Figure 1. It is important to understand how these approximations translate to the distance of $\mu_{\text{sample}}$ to $\mu_{\text{data}}$ or $\hat{\mu}_{\text{data}}$.

Some early works already deal with these questions. In [De Bortoli et al., 2021] the total variation distance between $\mu_{\text{sample}}$ and $\mu_{\text{data}}$ in bounded, whereas [Song et al., 2021a] derives bounds with

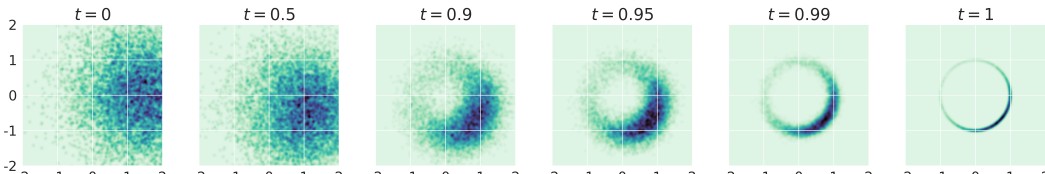

(a) Here $\mu_{\text{data}}$ is chosen as the uniform distribution on the unit sphere $\mathcal{S}^1$. We use the Brownian motion for the forward SDE. We can compute the exact score $\nabla \log p_t(x)$. We perturb it with the vector $v = (x = 0, y = -1)$ and define the approximation $s(x, t) = \nabla \log p_t(x) + v$. Furthermore, we purposely adopt a poor approximation $\mu_{\text{prior}} \not\approx p_1$ by setting $\mu_{\text{prior}} = \mathcal{N}(m, I)$, where $m = (x = 1.5, y = 0)$. We then run the reverse SDE (8). The figure shows heat maps of the intermediate distributions $q_t$ of the reverse SDE. At time $t = 1$ we reach $q_1 = \mu_{\text{sample}}$. We see that $\mu_{\text{sample}}$ is a distribution on $\mathcal{M} = \mathcal{S}^1$, albeit not the uniform one. Furthermore, we can observe how the errors in the initial conditions and the drift influence the skewed distribution $\mu_{\text{sample}}$. The initial conditions where chosen to have a to large $x$-coordinate on average, whereas the drift was chosen as to prefer lower $y$-coordinates. The distribution $\mu_{\text{sample}}$ is concentrated in areas with high $x$ and low $y$-coordinates.

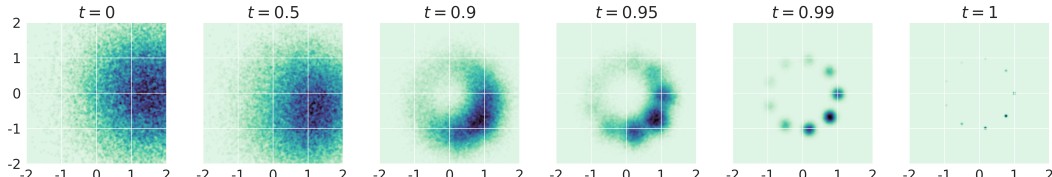

(b) The above experiment is repeated but with $\mu_{\text{data}}$ chosen to be the uniform distribution on $\mathcal{M} = \{x_i\}_{i=1}^9$, where the $x_i$ are 9 evenly spaced points on the unit sphere $\mathcal{S}^1$. Notice that while the approximation errors cause a non-uniform distribution the training examples, $\mu_{\text{sample}}$ is still supported solely on $\mathcal{M}$ and will not generate novel samples.

Figure 2: Perturbing $\nabla \log \hat{p}_t$.

respect to the KL-Divergence. The work [De Bortoli et al., 2021] furthermore derives a result similarly to the second part of Theorem 1, but also treating the errors that are introduced by discretizing the SDE. However, both of these works assume that the initial distribution $\mu_{\text{data}}$ is rather well behaved. In particular, it is assumed that $\mu_{\text{data}}(x) > 0$ for all $x$. We shortly discuss this assumption now.

Assuming that $\mu_{\text{data}}(x) > 0$ for any $x$ means that one postulates that every $x$ is a possible sample from $\mu_{\text{data}}$. For example, this implies that even if all samples $\{x_1, \ldots, x_n\}$ of $\mu_{\text{data}}$ consist of images of human faces, we say that $\mu_{\text{data}}$ can possibly also generate images of for example furniture, animals or pure white noise. Even though it might have low probability, any combinations of pixels is a possible sample from $\mu_{\text{data}}$. The assumption that $\mu_{\text{data}}$ is actually supported on some lower dimensional substructure $\mathcal{M}$ is well known under the name *manifold hypothesis*, see for example [Bengio et al., 2013, Pope et al., 2021]. In practice this means that $\mu_{\text{data}}(x) = 0$ for many $x$. This also leads to exploding scores $\nabla \log p_t$ as $t \to 0$ (see Section 5), a behaviour which also observed empirically and further underpins the relevance of the manifold assumption. We will from now on denote the support of $\mu_{\text{data}}$ as *data manifold* $\mathcal{M}$.

A fundamental question is then: What is the support of $\mu_{\text{sample}}$ and how does it compare to $\mathcal{M}$. This is interesting for multiple reasons. On the one hand, if we for example assume that $\mathcal{M}$ is the set of all images of faces, then the knowledge that $\mu_{\text{sample}}$ also has support $\mathcal{M}$ implies that, regardless of how close $\mu_{\text{sample}}$ actually is to $\mu_{\text{data}}$, it will at least always produce an image of a face. On the other hand, we can also compare the support of $\mu_{\text{sample}}$ to that of $\hat{\mu}_{\text{data}}$. The measures $\mu_{\text{sample}}$ and $\mu_{\text{data}}$ sharing their support translates to $\mu_{\text{sample}}$ memorizing the training data and not being able to generalize. Both of these are very valuable insights into the qualities of $\mu_{\text{sample}}$ on its own. Furthermore, statistical distances like the KL-Divergence or the total variation distance are only meaningful if the support of the measures overlap to some degree, as otherwise they will equal the maximum distance value.

If two measures have the same support, they are said to be *equivalent*. Our main result is the following:

**Main result.** *We identify conditions under which $\mu_{sample}$ is able to learn the data manifold, that is $supp(\mu_{sample}) = supp(\mu_{data})$. Applying these results to different settings, we find precise conditions*

*under which an SGM memorizes its training data or under which it is able to learn the right data manifold $\mathcal{M}$.*

*In particular we find that for SGMs to be able to generalize, the approximation error made when approximating the training drift $\nabla \log \hat{p}_t$ has to be unbounded.*

A first illustration of these results is given in Figure 2. We use a simple example, where we can perfectly evaluate the true drift $\nabla \log \pi_t$. We then choose an incorrect initial condition $\mu_{\text{prior}}$ which is far from $p_T$, and also add a constant error to $\nabla \log \pi_t(x)$. The initial measures $\pi_0$ are given as the uniform distribution on the unit sphere and the uniform distribution on 9 samples from the unit sphere in Figure 2a and Figure 2 respectively. We see that the final distribution of the reverse SDE, $\mu_{\text{sample}}$, is not the uniform distribution anymore. This is due to errors in the initial conditions and the drift. Nevertheless, $\mu_{\text{sample}}$ is still supported on the exact same subset as $\pi_0$.

When applying our main result to the empirical measure $\hat{\mu}_{\text{data}}$ one gets the following corollary, supplying precise conditions under which a SGM has memorized its training data.

**Corollary 1.** *Denote by $\hat{X}_t$ the forward SDE when started in the empirical measure $\pi_0 = \hat{\mu}_{data}$. Let $\int_0^T \|s_\theta(\hat{X}_t, t) - \nabla \log \hat{p}_t(\hat{X}_t) \mathrm{d}t$ be drift approximation error along a path of the forward SDE. For a given weighting function $w(t)$, the training objective of an SGM can be written as*

$$L_2 = \mathbb{E}_{\hat{X}}[\int_0^T w(t)\|s_\theta(\hat{X}_t, t) - \nabla \log \hat{p}_t(\hat{X}_t)\|^2 \mathrm{d}t], \tag{3}$$

*see Section 3. Simultaneously, if the exponential integral of the drift approximation error is integrable in the following sense:*

$$L_{\text{exp}} = \mathbb{E}_{\hat{X}}[\exp(\frac{\sigma}{2}\int_0^T \|s_\theta(\hat{X}_t, t) - \nabla \log \hat{p}_t(X_t)\|^2 \mathrm{d}t)] < \infty, \tag{4}$$

*the SGM has memorized its training data. Therefore, while training an SGM one should aim to minimize the mean squared error 3 while ensuring that the mean exponential error stays infinite, $L_{\text{exp}} = \infty$.*

In particular, if $\|s_\theta(\hat{X}_t, t) - \nabla \log \hat{p}_t(X_t)\|$ is bounded, then $L_{\text{exp}}$ is finite. Therefore, the the generalization capability of a SGM crucially depends on the training error being unbounded.

We now proceed as follows. In Section 2 we will summarize some of the most popular forward SDEs that are applied in SGMs. Then, in Section 3 we discuss how the drift approximation $s_\theta(x, t)$ for SGMs is trained in most implementations. We are ready to state our main results in Section 4. In Section 5 we show how the empirically observed drift explosion is related to the the manifold hypothesis. Most of the paper discusses the error in the drift approximation. But as we have discussed, we also have an error in the initial conditions. In Section 6 we will discuss how large the error in the initial conditions will be in practice.

## 2 Popular SDEs used in SGMs

In this section we introduce some of the most popular SDEs that are used when implementing SGMs. The first works on SGMs studied discrete forward and backward processes. Nevertheless, the transition kernels and algorithms proposed in those works can be seen as discretisations of some well-known SDEs. More recent works have studied this connection and state the algorithms in terms of SDEs ([Song et al., 2021b, Huang et al., 2021]).

**Brownian Motion:**   The works [Song and Ermon, 2019, 2020] can be seen as a discretization of the SDE

$$\mathrm{d}X_t = \sigma(t)\mathrm{d}W_t.$$

Denoting $h(t) = \int_0^t \sigma(s)\mathrm{d}s$, the solution to the above process can be explicitly stated as a time-changed Brownian motion, $X_t = W_{h(t)}$. The time-change can help in the implementation but does not alter the qualitative behaviour of the reverse SDE. In our following analysis we therefore set $\sigma(t) = 1$. Nevertheless, our results still hold for any positive $\sigma(t)$.

**Ornstein-Uhlenbeck Process:** The works [Sohl-Dickstein et al., 2015, Ho et al., 2020] can be seen as a discretization of

$$\mathrm{d}X_t = -\frac{1}{2}\alpha(t)X_t\mathrm{d}t + \sqrt{\alpha(t)}\mathrm{d}W_t,$$

which is an Ornstein-Uhlenbeck process. Again, the parameters $\alpha_t$ are a time-change and do not influence the properties that we are investigating in this paper. Therefore, to simplify notation, we again set $\alpha(t) = 1$.

**Critically Damped Langevin Dynamics (CLD):** The work [Dockhorn et al., 2021] studies a second order SDE. Here artificial velocity coordinates $V_t$ are introduced and the system under consideration is

$$
\begin{aligned}
\mathrm{d}X_t &= V_t, \\
\mathrm{d}V_t &= -X_t - 2V_t + 2\mathrm{d}W_t,
\end{aligned}
$$

where $X_0 \sim p_{\text{data}}$, $V_0 \sim \mathcal{N}(0, I)$. For generation one runs the reverse SDE in $X_t$ and $V_t$ but discards the $V$ coordinate at the end. The work [Dockhorn et al., 2021] also includes the parameters $M$ and $\gamma$. We set both to 1 as they do in their numerical experiments.

## 3 Score approximation with a finite number of samples

We now quickly discuss how the neural network is trained to approximate $\nabla \log p_t$ and what implications this has. The score $\nabla \log p_t$ is approximated by minimizing a variant of

$$L(\theta) = \int_0^T w(t) \, \mathbb{E}_{p_t(x)}[\|\nabla \log p_t(x) - s_\theta(x,t)\|^2]\mathrm{d}t. \tag{5}$$

for some weighting function $w$. The optimization is done via score matching techniques (see [Hyvärinen and Dayan, 2005, Vincent, 2011, Song et al., 2020]). However, the above expectation depends on $p_t$, which we cannot evaluate since it depends on $p_0 = \mu_{\text{data}}$. Nevertheless, we can evaluate the approximation $\hat{p}_t$,

$$\hat{p}_t(x) = \mathbb{E}_{\hat{\mu}_{\text{data}}(x_0)}[p_{t|0}(x|x_0)] = \frac{1}{N}\sum_{i=1}^N p_{t|0}(x|x^i) \approx p_t(x) = \mathbb{E}_{\mu_{\text{data}}}[p_{t|0}(x|x_0)]. \tag{6}$$

The surrogate loss

$$\hat{L}(\theta) = \int_0^T w(t) \, \mathbb{E}_{\hat{p}_t(x)}[\|\nabla \log \hat{p}_t(x) - s_\theta(x,t)\|^2]\mathrm{d}t, \tag{7}$$

can be evaluated and is used for training. The equation (7) is equivalent to (3) since $\hat{X}_t$ has distribution $\hat{p}_t$. If we would minimize this loss perfectly, then $s_\theta(x,t)$ would be equal to $\nabla \log \hat{p}_t$. The reverse SDE started in an appropriate initial condition with drift $\nabla \log \hat{p}_t$ however, will produce samples from $\hat{\mu}_{\text{data}}$, which are training examples. This raises the question in which way or to which degree the loss should actually be minimized.

## 4 Effects of the approximations

This section contains our main results. We first state the assumptions we have to make and then the Theorems.

### 4.1 Error in the initial condition

The following assumption is needed for the reverse SDE to be defined even in the case when the initial distribution $\pi_0$ has a degenerate support. In Lemma 1 we will show that all the SDEs from Section 2 satisfy the Assumption.

**Assumption 1.** *There is a constant $C$ such that*

    *(i) $\beta$ is globally Lipschitz, i.e.$\|\beta(x) - \beta(y)\| \leq C\|x - y\|$.*

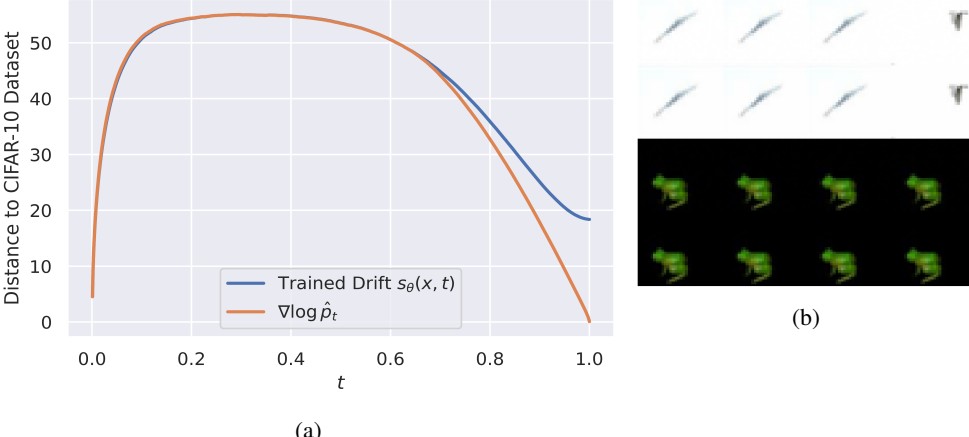

(a)

(b)

Figure 3: (a): Both lines correspond to the same experiment for different drifts in the reverse SDE. For both lines we started $N = 1000$ paths in the zero vector in $\mathbb{R}^{32 \times 32 \times 3}$. For the blue line we used the pretrained CIFAR-10 DDPM++ model from Song et al. [2021b], whereas for the orange line we used the true drift $\nabla \log \hat{p}_t$, which is a mixture of $50000$ Gaussians, one for each training example in CIFAR-10. We then saved the distance from $Y_t$ to the CIFAR-10 training examples, by calculating the distance to the closest example. Above we plot the average distance. We see, that while the reverse SDE run with $\hat{p}_t$ will have a distance of 0 to the training examples in the end, the SDE with the pretrained drift keeps some distance to the training examples and therefore produces novel images. (b): We evaluate $\nabla \log \hat{p}_t$ as in (a). We do the analogous experiment to Figure 2b on CIFAR-10 and perturb the empirical drift $\nabla \log \hat{p}_t$ with a constant error vector. The first row shows the samples generated by adding the constant error vector $e(x, t) = (1, 1, \dots, 1) \in \mathbb{R}^{32 \times 32 \times 3}$ to $\nabla \log \hat{p}_t$. In the second row we searched for the closest image in the CIFAR-10 dataset (with respect to the Euclidean 2-distance on $\mathbb{R}^{32 \times 32 \times 3}$) and plotted it. We see that all the sampled images are nearly equal to a corresponding image in CIFAR-10. The distance of the images to their closest image in CIFAR-10 is around 0.07 for all plotted images. Similar to Figure 2, we can observe the effect of adding the one-vector. The sample distribution $\mu_{\text{sample}}$ got skewed to prefer images that have high pixel values. This corresponds to samples which are mostly white for the human eye. In the third and forth row we repeat the experiment of the first and second row, but add the negative one-vector $e(x, t) = (-1, -1, \dots, -1)$ and get black images.

(ii) $\beta$ grows at most linearly, i.e. $\|\beta(x)\| \le C(1 + \|x\|)$.

(iii) $X_t$ has a density $\pi_t \in C^1$ for every $t > 0$ and $\int_{t_0}^1 \int_{\|x\| < R} |\pi_t(x)|^2 + \|\nabla_x \pi_t(x)\|^2 \mathrm{d}x\mathrm{d}t < \infty$ for any $R > 0$ and $0 < t_0 \le T$.

*Furthermore, for each $S \in (0, T)$ and all $x, y$ for which $\|x\|, \|y\| \le N$ there is a constant $C_{S,N}$ such that*

(iv) $\nabla \log \pi_t$ *is locally Lipschitz,* $\|\nabla \log \pi_t(x) - \nabla \log \pi_t(y)\| \le C_{S,N} \|x - y\|$ *for all $t \in [S, T]$.*

Conditions $(i)$-$(iii)$ are technical conditions on the forward SDE. They ensure that if we run a solution to the forward SDE, $X_t$, backwards in time, then $X_t^R := X_{T-t}$ will be a solution to the reverse SDE (2) on $[0, T]$. The last condition then ensures that the solutions to the reverse SDE are unique, therefore we will be able to transmit the properties of $X^R$ to any other solution $Y_t$ of (2). The following result shows that Assumption 1 can be expected to hold in practice. To simplify the calculations we assume that the data manifold $\mathcal{M} = \text{supp}(\mu_{\text{data}})$ is contained in a ball of diameter $M$. This is a natural assumption for many data sets. Nevertheless, we note that this assumption could be weakened by additional technical effort.

**Lemma 1.** *Assume that the data manifold $\mathcal{M}$ is contained in a ball of radius $M$. Then all the methods introduced in Section 2 fulfil Assumption 1.*

We are now ready to state our first main result,

**Theorem 1.** *Denote by $\pi_t$ the marginals of the forward SDE started in $\pi_0$. Assume that Assumption 1 holds and that $\mu_{prior}$ is absolutely continuous with respect to $\pi_T$. Then the following hold.*

    *(i) Let $Y_t$ be a solution to (2) on $[0, T)$. The limit $Y_T := \lim_{t \to T} Y_T$ exists almost surely. We refer to its distribution as $\mu_{sample}$. The distribution $\mu_{sample}$ is absolutely continuous with respect to $\mu_{data}$. If $\pi_T$ and $\mu_{prior}$ are equivalent, then so are $\mu_{sample}$ and $\pi_0$.*

    *(ii) Furthermore, for any $f$-divergence $D_f$,*

$$D_f(\mu_{sample}|\pi_0) \leq D_f(\mu_{prior}|\pi_T) \quad \text{and} \quad D_f(\pi_0|\mu_{sample}) \leq D_f(\pi_T|\mu_{prior}).$$

Applying the above theorem with $\pi_T = p_T$ tells us something about the equivalence and distance between $\mu_{sample}$ and $\mu_{data}$, since $\pi_0 = p_0 = \mu_{data}$. Applying the theorem with $\hat{p}_t$ tells us something about the generalization capabilities of SGMs.

The measures $\pi_T$ and $\mu_{prior}$ are normally both supported on all of $\mathbb{R}^d$ and therefore equivalent. Since Assumption 1 also holds in most situations (see Lemma 1), the requirements for Theorem 1 are satisfied in practice. From item $(i)$ with $\pi_t = \hat{p}_t$ we can conclude that the error we make in the initial conditions is not responsible for the generalization capacities of SGMs.

The second item then shows that the $f$-divergences between $\mu_{sample}$ and $\mu_{data}$ are bounded by the $f$-divergences of $\mu_{prior}$ to $p_T$. The total variation distance and the KL-Divergence are both special cases of $f$-divergences.

## 4.2 Error in the drift

Given a forward SDE with marginals $\pi_t$ and an approximation $s(x, t)$ to $\nabla \log \pi_t$, we define the reverse SDE for the approximation $s$ as

$$
\begin{aligned}
\mathrm{d}\tilde{Y}_t &= -\beta(\tilde{Y}_t)\mathrm{d}t + \sigma\sigma^T s(\tilde{Y}_t, t)\mathrm{d}t + \sigma\mathrm{d}B_t, \\
\tilde{Y}_0 &\sim q_0.
\end{aligned}
\tag{8}
$$

**Assumption 2.** *We assume that the reverse SDE $\tilde{Y}_t$ has a solution on $[0, T)$. For $t < T$, we define the Girsanov weights*

$$Z_t = \exp\left( \int_0^t \sigma^T(s(\tilde{Y}_t, t) - \nabla \log \pi_t(\tilde{Y}_t)) \cdot \mathrm{d}B_s - \frac{1}{2} \int_0^t \|\sigma^T(s(\tilde{Y}_t, t) - \nabla \log \pi_t(\tilde{Y}_t))\|^2 \mathrm{d}s \right)$$

$$\tag{9}$$

*and assume that the $Z_t$ are a uniformly integrable martingale.*

We shortly discuss this assumption. The assumption that $Z_t$ is a martingale is equivalent to the expectation of $Z_t$ being equal to 1 for all $t$. The assumption that it is uniformly integrable is more technical (see Appendix A.2), but is fulfilled for example if for some $p > 1$, $\mathbb{E}[|Z_t|^p] < \infty$ for all $t$. For example, if the $Z_t$ have bounded variance, then they are uniformly integrable.

A condition that ensures that $Z_t$ is both, a martingale and uniformly integrable, is given by *Novikov's condition* Novikov [1980]. It states, that if

$$N_T = \mathbb{E}_{\tilde{Y}}\left[ \exp\left( \frac{1}{2} \int_0^T \|\sigma^T(s(\tilde{Y}_t, t) - \nabla \log \pi_t(\tilde{Y}_t))\|^2 \mathrm{d}s \right) \right] < \infty, \tag{10}$$

then $Z_t$ is a uniformly integrable martingale.

Using Assumption 2 we can now state

**Theorem 2.** *Assume that Assumption 2 holds. Assume furthermore that Assumption 1 holds with $\nabla \log \pi_t$ replaced by $s(x, t)$.*

*Then $\tilde{Y}_T = \lim_{t \to T} \tilde{Y}_t$ is well defined. Moreover, its distribution is equivalent to the distribution of $Y_T$. In particular, if $\|s(x, t) - \nabla \log \hat{p}_t\|$ is bounded, then Assumption 2 holds, and the SGM has memorized its training data.*

Putting Theorem 1 and Theorem 2 together, we can conclude that, if both Assumptions hold, $\mu_{sample}$ is equivalent to $\pi_0$. Therefore, if Assumption 2 holds for $\pi_t = p_t$, we have a positive statement and know that $\mu_{sample}$ will have the exact same support as $\mu_{data}$, i.e. it has learned the data manifold $\mathcal{M}$.

If the Assumption would hold for $\pi_t = \hat{p}_t$, we would however just memorize the training data, see Figure 2b or Figure 3b for a visualization. However, empirically it has been shown that SGMs are able to create novel samples (see, Figure 3a or Dhariwal and Nichol [2021]). Therefore, we can deduce that Assumption 2 is violated in practice.

We evaluated N from (10) on CIFAR-10, once for the difference between the $s_\theta(x, t)$ from Song et al. [2021b] and $\nabla \log \hat{p}_t$, and once by just using a perturbed drift with a constant error, $s(x, t) = \nabla \log \hat{p}_t + \frac{1}{2}(1, 1, \ldots, 1)$, see Figure 4. The reverse SDE using the drift $\nabla \log \hat{p}_t + \frac{1}{2}(1, 1, \ldots, 1)$, is equivalent to $\hat{\mu}_{\text{data}}$, as we know from Theorem 2 and have also already observed in Figure 3b. Figure 4 confirms that $Z_t$ is indeed a uniformly integrable martingale and therefore fulfils Assumption 2.

Corollary 1 can be deduced by exchanging the roles of $\tilde{Y}_t$ and $Y_t$. We then get an equivalent condition to Assumption 2, which is

$$\tilde{N}_T = \mathbb{E}_Y \left[ \exp \left( \frac{1}{2} \int_0^T \|\sigma^T(s(Y_t, t) - \nabla \log \pi_t(Y_t))\|^2 \mathrm{d}s \right) \right] < \infty,$$

where the expectation is now taken over the reverse SDE with the correct drift $\nabla \log \pi_t$ instead of the approximate drift $s_\theta$. However, $Y$ is just the time reversal of $X$, therefore we can also write the above expectation over $X_t$ instead of $Y_t$. If we treat the case where we start $X_t$ in the empirical measure $\hat{\mu}_{\text{data}}$, $\pi_t$ will be equal to $\pi_t = \hat{p}_t$ by definition and

$$\mathbb{E}_{\hat{X}} \left[ \exp \left( \frac{\sigma^2}{2} \int_0^T \|\sigma^T(s(\hat{X}_t, t) - \nabla \log \hat{p}_t(\hat{X}_t))\|^2 \mathrm{d}s \right) \right] < \infty$$

is a sufficient condition for $\hat{\mu}_{\text{data}}$ and $\mu_{\text{sample}}$ having the same support. We have here assumed that Theorem 1 can be applied. However, this can be assumed in practice, see the discussion after Theorem 1.

In future work we believe that further understanding the characteristics of $Z_t$, how they relate to the minimization of $L(\theta)$ and generalization is crucial to the understanding of SGMs and their empirical success. Recent works also study the question on how the neural network architecture and parametrization is related to the boundedness of the output Kim et al. [2022]. The relationship between these choices and the properties of $Z_t$ is also an interesting research avenue. Lastly, the distribution of $Z_t$ is very heavy-tailed. Most of the samples are very small with a few extremely large samples in between. Therefore one needs many samples from $Z_t$ to understand its characteristics. Finding robust estimators for $Z_t$ or its expectation could help their usage in the training or evaluation procedure.

## 5 Drift Explosion under manifold hypothesis

In practice it is often observed that the drift of the reverse SDE explodes as $t \to T$. see for example Kim et al. [2022, Section 3.1]. We now show how this observed behaviour is related to the manifold hypothesis.

First, we note that all SDEs in Section 2 are linear SDEs. Therefore, their transition kernels are Gaussian ([Pavliotis, 2014, Section 3.7]):

$$\pi_t(X_t = x | X_0 = x_0) = \mathcal{N}(x; m_t(x_0), \Sigma_t).$$

The explicit form of $m_t$ and $\Sigma_t$ differ for each of the SDEs and can be found in Appendix C.1. We remark that $\Sigma_t$ does not depend on the initial condition $x_0$. The transition kernel above is the distribution of the SDE started in a single point $x_0$. Since we start the SDE in $\mu_{\text{data}}$ we need to average over $\mu_{\text{data}}$ to get the marginal at time $t$:

$$p_t(x) = \int_{\mathbb{R}^d} \mathcal{N}(x; m_t(x_0), \Sigma_t) \mu_{\text{data}}(x_0) \mathrm{d}x_0. \tag{11}$$

We can also compute the additional drift in the reverse SDE (see Appendix C.2),

$$\nabla \log p_t(x) = \frac{\nabla p_t(x)}{p_t(x)} = \Sigma_t^{-1}(x - \mathbb{E}[m_t(X_0) | X_t = x]). \tag{12}$$

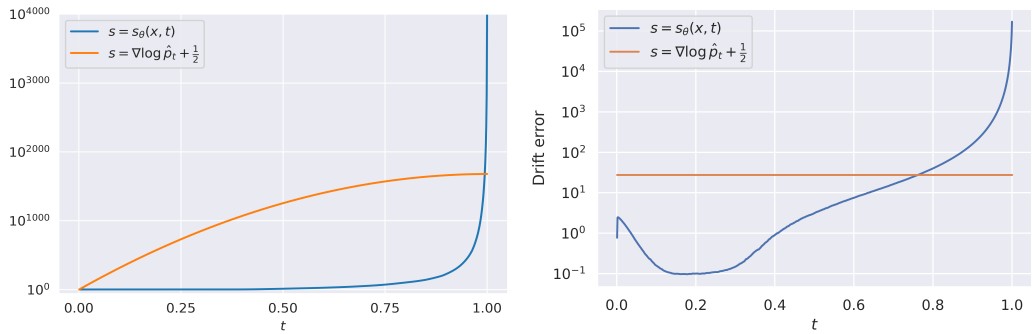

Figure 4: Left: We simulated the reverse SDE on CIFAR-10, once with the pretrained CIFAR-10 DDPM++ model $s_\theta$ from Song et al. [2021b] and once with a perturbed drift $s(x,t) = \nabla \log \hat{p}_t + \frac{1}{2}(1,1,\ldots,1)$. We then evaluated the integral (10) numerically for varying $t = T$. For the perturbed drift, the integral does not seem to explode as $t \to 1$, implying that $Z_t$ is a martingale. We see that for the DDPM++ drift, the integral explodes, therefore we can not infer that $Z_t$ is a martingale. We used $N = 12000$ simulations from both of the SDEs to generate this plot. Right: We again ran the two SDEs with the drifts as in the left Figure. This time, we measured the average distance to the empirical drift $\|s(\hat{Y}_t,t) - \nabla \log \hat{p}_t(\hat{Y}_t)\|$ along a path of the reverse SDE. We repeated the experiment $N = 2560$ times and plotted the mean distance. For the constant perturbation we also of course get a constant distance. The distance of the true drift to $\nabla \log \hat{p}_t$ is initially very small but explodes as $t \to 1$. From our results we know that this explosion is necessary for the SGM to generalize.

We now want to evaluate $\nabla \log p_t$ along a typical path of $Y_t$, i.e. we are interested in $\mathbb{E}[\nabla \log p_t(Y_t)]$. The distribution of $Y_t$ however depends on the drift approximation $s_\theta$ we use. For this calculation we will assume that we are able to run the reverse SDE with the true drift $s_\theta(x,t) = p_t(x)$. Then however, since $Y_t$ is then just $X_t$ run backwards, they have the same distributions and we can calculate

$$\mathbb{E}[\|\nabla \log p_t(Y_t)\|] = \mathbb{E}[\|\nabla \log p_t(X_t)\|] = \Sigma_t^{-1}\mathbb{E}[\|X_t - \mathbb{E}[m_t(X_0)|X_t = y]\|].$$

If the manifold $\mathcal{M}$ is not to badly behaved, we can expect that for small $t$ and almost all $x$, $\mathbb{E}[X_0|X_t = x]$ to be very close to the data manifold $\mathcal{M}$. Especially, $\|X_t - \mathbb{E}[X_0|X_t = x]\|$ will be larger than $\text{dist}(X_t, \mathcal{M})$ in that case. However, the distribution of $X_t$ can be represented as $m_t(X_0) + \sqrt{\Sigma_t}\xi$, where $\xi$ has a standard normal distribution. For the SDEs we treated in Section 2, $m_t(X_t)$ is either equal or very close to $m_t(X_t) = X_t$ for small values of $t$. Finally, if we assume that $\mathcal{M}$ is a subset of relatively low dimension in a high dimensional space, we can expect that with very high probability $\sqrt{\Sigma_t}\xi$ points away from the data manifold. Therefore the distance of $X_t$ to $\mathcal{M}$ can be approximated by $\sqrt{\Sigma_t}\xi$. Putting these approximations together we can calculate

$$\mathbb{E}[\|X_t - \mathbb{E}[m_t(X_0)|X_t = y]\|] \gtrsim \mathbb{E}[\text{dist}(X_t, \mathcal{M})] \gtrsim \|\Sigma_t\|^{1/2}\mathbb{E}[\|\xi\|] \approx \|\Sigma_t\|^{1/2}\sqrt{d},$$

where $d$ is the dimension of the data space in which the samples $x_i$ lie. Therefore we can conclude that

$$\|\nabla \log p_t(Y_t)\| \gtrsim \frac{d^{1/2}}{\|\Sigma_t\|^{1/2}}.$$

For the Brownian motion for example, $\Sigma_t = t$ and therefore the right hand side scales like $\frac{1}{\sqrt{t}}$. If $\Sigma_t$ is the covariance of $p_t$, then we can expect $\nabla \log p_t(Y_t)$ to be of order $\frac{1}{\|\Sigma_t\|^{1/2}}$. Furthermore, $\nabla \log p_t(Y_t)$ will point towards the data manifold for small $t$. The drift $\nabla \log p_t(Y_t)$ then acts like a *support matching force*, where the force grows to infinity as $t \to 0$, absorbing all the SDE paths onto the manifold.

## 6   Distance from $p_T$ to $\mu_{\text{prior}}$

We have seen in Theorem 1 that the distance between $\mu_{\text{sample}}$ and $\mu_{\text{data}}$ is directly related to the distance between $p_T$ and $\mu_{\text{prior}}$ if we neglect the errors made in the approximation of the drift. For both, the OU-Process and the CLD, there are plenty of results on the distance of $p_t$ to $\mathcal{N}(0, I_d)$. In general, one can expect this distance to grown exponentially in time $t$.

The Brownian motion however does not converge to a stationary distribution and therefore one has to choose a different $\mu_{\text{prior}}^T$ for each $T$ to approximate $p_T$. In practice, one normally chooses a normal distribution $p_T = \mathcal{N}(m_t, C_t)$ (see [Song et al., 2021b, Appendix C]). The following Lemma is derives the optimal values for $m_t$ and $C_t$.

**Lemma 2.** *Let $p_T$ be the $T$-time marginal of the Brownian motion process $X_t$ of Section 2. The following minimization problem*

$$\min_{m,C} KL(p_T \mid \mathcal{N}(m, C))$$

*is minimized by $m_T = \mathbb{E}[\mu_{data}]$ and $C_T = \text{Cov}[\mu_{data}] + T I_d$. If we restrict the covariance to be a multiple of the identity matrix, the problem is solved by choosing $m$ as above and $c$ as $c = \mathbb{E}[\|X_t - m\|^2] = trace(C_T)$.*

This result is a slight variation on the well known fact that the $KL$-projection in the second argument matches its moments. We prove it in Appendix E.2. The following result shows that the distance between $p_T$ and $\mu_{\text{prior}}^T$ decreases with time and also gives a rate. It justifies using the Brownian motion and a normal prior distribution for SGMs.

**Lemma 3.** *Let $p_t$ be the time $t$-marginal of a Brownian motion with initial condition $\mu_{data}$. Denote by $c_i, i = 1, \ldots, d$, the eigenvalues of $\text{Cov}(\mu_{data})$. Let $\mu_{prior}^T$ be the normal distribution with mean $m_T = \mathbb{E}[\mu_{data}]$ and covariance $C_T = \text{Cov}[\mu_{data}] + T I_d$. Then*

$$KL(p_T|\mu_{prior}^T) \leq \frac{1}{2} \log \left( \frac{\prod_{i=1}^d (c_i + T)}{T^d} \right).$$

The proof can be found in Appendix E.2.

## 7 Broader impact

The results deepen the understanding of score-based generative models. As such, they can be seen as a step towards improving the quality of generative models. Therefore the possible negative societal impacts are the same ones that apply to generative modelling in general. First, generative models can be used to create synthetic data that is hard to distinguish from real data (for example images or videos), see [Mirsky and Lee, 2021]. Second, generative models can learn and reproduce biases that are prevalent in the training data ([Esser et al., 2020]). Last, depending on the application, generative models might be used to do creative work that was previously done by humans.

## 8 Conclusion

We conducted a theoretical study of some properties of SGMs. We found explicit conditions under which the sample measure $\mu_{\text{sample}}$ is equivalent to the true data generating distribution $\mu_{\text{data}}$. Under these conditions we can guarantee, that the SGM generates samples that could also be samples from $\mu_{\text{data}}$. Furthermore, each sample that can be generated by $\mu_{\text{data}}$ also has positive probability under $\mu_{\text{sample}}$, meaning that the full support is covered.

Since one can not actually access the full support of $\mu_{\text{data}}$, but only a finite number of training examples $\{x_i\}_{i=1}^N$, our results can be applied to find conditions under which the SGM memorizes its training data. We believe that this observation provides a first step towards understanding the generalization capabilities of SGMs.

## 9 Funding

The author has been partially supported by Deutsche Forschungsgemeinschaft (DFG) - Project-ID 318763901 - SFB1294.

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
