# A Stochastic prerequisites

In this section we give a formal introduction to some of the concepts used in this work. For a more rigorous treatment, see for example [Klenke, 2013] or [Karatzas and Shreve, 2012].

## A.1 Equivalence of measures / Girsanov Theorem

First we define absolute continuity of measures. Let $\mu$ and $\nu$ be two measures on $(\Omega, \mathcal{F})$, where $\mathcal{F}$ is a $\sigma$-algebra.

**Definition 1.** *We say that $\mu$ is absolutely continuous with respect to $\nu$ if $\mu(A) = 0$ for any $A \in \mathcal{F}$ such that $\nu(A) = 0$. We also denote this by $\mu \ll \nu$.*

Two measures $\mu$ and $\nu$ are equivalent if $\mu \ll \nu$ and $\nu \ll \mu$. Loosely speaking, we can say that $\mu \ll \nu$ if the support of $\mu$ is contained in the support of $\nu$ and they are equivalent if they share the same support.

The Radon-Nikodym theorem tells us that if $\mu \ll \nu$, then under mild conditions there exists a density $\frac{\mathrm{d}\mu}{\mathrm{d}\nu} : \Omega \to \mathbb{R}$ such that $\mu(A) = \int_A \frac{\mathrm{d}\mu}{\mathrm{d}\nu}(\omega)\mathrm{d}\nu(\omega)$. Therefore, we can obtain $\mu$ through a reweighting of $\nu$. One specific instance of this is the Girsanov Theorem. Assume we are given the solutions to two SDEs in $\mathbb{R}^d$,

$$\mathrm{d}Y_t = b(t, Y_t)\mathrm{d}t + \sigma(t, Y_t)\mathrm{d}W_t \tag{13}$$

and

$$\mathrm{d}\tilde{Y}_t = b(t, \tilde{Y}_t)\mathrm{d}t + \sigma(t, \tilde{Y}_t)e(t, \tilde{Y}_t)\mathrm{d}t + \sigma(t, \tilde{Y}_t)\mathrm{d}B_t. \tag{14}$$

Both of these induce a measure on the space of continuous functions $\Omega = C([0, T], \mathbb{R}^d)$. We denote them by $\mathbb{P}$ and $\tilde{\mathbb{P}}$ respectively. Then the Girsanov Theorem equips us with conditions under which the measures $\mathbb{P}$ and $\tilde{\mathbb{P}}$ are equivalent. Furthermore, in case of equivalence we get a formula for the density of $\tilde{\mathbb{P}}$ with respect to $\mathbb{P}$. The relative density is given as

$$Z_T = \exp\left(\int_0^T e(s, Y_s)\mathrm{d}W_s - \frac{1}{2}\int_0^T \|e(s, Y_s)\|^2\mathrm{d}s\right).$$

For a full statement of the Girsanov Theorem and under which conditions it holds, see [Karatzas and Shreve, 2012, Section 3.5].

## A.2 Uniform integrability

Since we are treating the case where the drift explodes as $t \to T$ we end up with densities

$$Z_t = \exp\left(\int_0^t e(s, Y_s)\mathrm{d}W_s - \frac{1}{2}\int_0^t \|e(s, Y_s)\|^2\mathrm{d}s\right). \tag{15}$$

on $C([0, t], \mathbb{R}^d)$, but not with a density on $C([0, T], \mathbb{R}^d)$. Uniform integrability is exactly the condition one needs to extend these local densities.

**Definition 2.** *A family $\{X_\alpha\}$ of random variables is called uniformly integrable if*

$$\sup_\alpha \mathbb{E}\left[|X_\alpha| \, 1_{\{|X_\alpha| > s\}}\right] \to 0$$

*as $s \to \infty$.*

In the proof of Theorem 2 we implicitly use the following two results which we here state as a lemma. The filtration $\mathcal{F}_t$ is defined as in the proof of Theorem 2.

**Lemma 4.** *Assume the $Z_t$ in (15) form a uniformly integrable martingale on $[0, T)$. Then,*

- *the limit $\lim_{t \to T} Z_t$ exists in $L^1$. We denote this limit by $Z$.*

- *Furthermore, $\tilde{\mathbb{P}}$ is absolutely continuous with respect to $\mathbb{P}$ on $\mathcal{F} = \sigma(\cup_{t < T}\mathcal{F}_t)$ with density $Z$.*

*Proof.* Both of these results are standard. The first one can for example be found in [Karatzas and Shreve, 2012, Section 1.3.B]. For the second one we compute that for any $A \in \mathcal{F}_s$,

$$\mathbb{E}_{\mathbb{P}}[1_A Z] = \mathbb{E}_{\mathbb{P}}[1_A \lim_{t \to T} Z_t] = \lim_{t \to T} \mathbb{E}_{\mathbb{P}}[1_A Z_t] = \mathbb{E}_{\mathbb{P}}[1_A Z_s] = \mathbb{E}_{\tilde{\mathbb{P}}}[1_A],$$

where we used $L^1$ convergence in the second equality and the martingale property of $Z_s$ in the third equality. Therefore $Z$ is a density of $\tilde{\mathbb{P}}$ with respect to $\mathbb{P}$ on each $\mathcal{F}_s$ for $s < T$. Therefore $Z$ is also a density of $\tilde{\mathbb{P}}$ with respect to $\mathbb{P}$ on $\mathcal{F}$ which concludes the proof. $\qquad\square$

# B  Numerics

All numerical experiments can be run on a consumer grade computer within a few minutes.

## B.1  Figure 1

We first discuss the top left figure of Figure 1. We set $p_0 = \mu_{\text{data}}$ to a mixture of two Gaussian $\mathcal{N}(-2, \frac{1}{100})$ and $\mathcal{N}(2, \frac{1}{100})$ with weights $w_1 = \frac{1}{3}$ and $w_2 = \frac{2}{3}$ respectively. Then, we draw $N = 5\,000\,000$ samples from $\mu_{\text{data}}$, denoted by $Y_0^n$, $n = 1, \dots, N$. An Euler-Maruyama discretization of the Brownian motion propagates these samples from time $t = 0$ to $t = 1$ by

$$X_{i+1}^n = X_i^n + \sqrt{dt} Z_i^n,$$

where $Z_n^i \sim \mathcal{N}(0, 1)$ are i.i.d. random variables, independent of $X_m^j$ for $m \leq n$ and $j = 1, \dots, N$. The time index $i$ runs from 0 to $I = 2000$ and $dt$ is set to $dt = \frac{1}{I}$. The initial samples $\{X_0^n\}_{n=1}^N$ are used to create the left line plot of $p_0$ and the final samples $\{X_I^n\}_{n=1}^N$ are used to create the right line plot of $p_1$ using kernel density estimation. The $\{X_i^n\}_{n=1}^N$ are approximate samples from $p_{i/I}$. Therefore, we create histograms using $\{X_i^n\}_{n=1}^N$ to approximate $p_{i/I}$. The height of the histogram bars corresponds to the square root of the colour intensity in the heat map. The horizontal axis in the heat map stands for the time $t$, whereas the vertical axis stands for the position $x$. At location $(t, x)$ we plot an estimate of $\sqrt{p_t(x)}$. We apply the square root since it improves the contrast in areas where $p_t(x)$ is close to 0 and makes it more visible where $p_t(x) > 0$ to the observer.

For the bottom left figure we show the same plots, just for the reverse SDE (8) instead of the forward SDE. Since the initial distribution is a Gaussian mixture we can exactly calculate $p_t$ using

$$p_t(x) = w_1 \mathcal{N}(x; m_1, s_1^2 + t) + w_2 \mathcal{N}(x; m_2, s_2^2 + t), \tag{16}$$

where we use $\mathcal{N}(x; m, v)$ for the probability density function of a normal distribution with mean $m$ and variance $v$, evaluated at $x$. With the above expression of $p_t$ one could compute an analytical representation of $\nabla \log p_t$. We use automatic differentiation instead. The reverse SDE (8) is simulated with a disturbance $e(x, t) = 1$ and initial condition $q_0 = \mu_{\text{prior}} = \mathcal{N}(0, 1)$. The Euler-Maruyama method is run with the same step size $dt = \frac{1}{I}$. More precisely, the one step transition kernel of the discretized reverse SDE is

$$Y_{i+1}^n = Y_i^n + dt \left(\nabla \log p_{1-\frac{i}{I}}(Y_i^n) + 1\right) + \sqrt{dt} \, \tilde{Z}_i^n, \tag{17}$$

where $\tilde{Z}_n^i \sim \mathcal{N}(0, 1)$ are i.i.d. random variables, independent of $Y_m^j$ for $m \leq n$ and $j = 1, \dots, N$. The plots are created in the same way as for the upper left plot, except that we reverse the time axis to plot $p_t$ and $q_{1-t}$ directly underneath each other.

On the right side we plot the same kernel density estimates already plotted on the left side as $\mu_{\text{sample}}$ and $\mu_{\text{data}}$ into the same plot for comparison.

## B.2  Figure 2b and 2a

Figure 2b is created by setting $\mu_{\text{data}}$ to be the uniform distribution on $M = 9$ equally spaced samples $\{x_i\}_{i=1}^M$ on the unit sphere $\mathcal{S}^1$. This can also be viewed as a Gaussian mixture with 9 components where each component having mean $x_i$ and variance 0. Therefore, we can again explicitly calculate $p_t$ for $t > 0$ as in (16),

$$p_t(y) = \frac{1}{M} \sum_{i=1}^{M} \mathcal{N}(y; x_i, t).$$

|                  | $m_t(z_0)$      | $\Sigma_t$              |
|------------------|-----------------|-------------------------|
| Brownian Motion  | $z_0$           | $tI_d$                  |
| OU-Process       | $\exp(-t)z_0$   | $(1 - \exp(-2t))I_d$    |

Table 1: The mean and covariance of the Gaussian transition kernels of the Brownian Motion and the Ornstein-Uhlenbeck process SDEs.

The score $\nabla \log p_t(y)$ is evaluated using automatic differentiation. The reverse SDE 8 is simulated with $q_0 = \mathcal{N}((x = -1.5, y = 0), I_2)$ and $e((x, y), t) = (x = 0, y = -1)$. For the numerical simulation, we again use the Euler-Maruyama scheme. We use a step width of $dt = \frac{0.9}{1000} \approx \frac{1}{1000}$ for $t = [0, 0.9]$, $dt = \frac{0.09}{1000} \approx \frac{1}{10\,000}$ for $t \in [0.9, 0.99]$ and of $dt = \frac{1}{100\,000}$ for $t \in [0.99, 1]$. For a simulation of $N = 50\,000$ paths of the reverse SDE, we start by drawing $Y_0^n = A_n + Z_n$, where $A_n$ are i.i.d. uniformly distributed on $\{x_i\}_{i=1}^M$ and $Z_n \sim \mathcal{N}(0, 1)$ i.i.d.. The $\{A_n\}$ and $\{Z_n\}$ are also independent from each other. We then propagate the $Y_0^n$ similarly to (17), except that we use a different values for $dt$ depending on $t$. This leads to approximate samples $Y_t^i$ from $q_t$. At the displayed times $t$ we plot the function

$$h_t(x) = \sum_{i=1}^N k(x, Y_t^i),$$

where $k$ is an unnormalized Gaussian kernel with a very small bandwidth parameter,

$$k(x, y) = \exp(-1000\|x - y\|^2).$$

Normalizing $h_t$ gives us a density estimate of $q_t$. We plot these estimates as heat maps for different values of $t$.

For Figure 2a we follow the same steps as for Figure 2b, except that $\mu_{\text{data}}$ is set to the uniform distribution over $M = 256$ evenly spaced samples from the unit sphere $\mathcal{S}^1$.

### B.3 Figure 3

For Figure 3 we used the DDPM++ model from the Github repository for the paper from Song et al. [2021b]. Then we evaluated the true score using (6), where the sum runs through the $N = 50000$ training examples of CIFAR-10, Krizhevsky et al. [2009]. A similar experiment using the true marginals $\hat{\mu}_{\text{data}}$ has also been conducted in Peluchetti [2021].

## C  Studying the forward Densities

### C.1  Transition kernels

The transition kernels $p(z_0, \cdot)$ for the SDEs from Section 2 are of the from

$$q_0(z_0, \cdot) = \mathcal{N}(m_t(z_0), \Sigma_t),$$

where $m_t$ and $\Sigma_t$ are given in Table 1 for the Brownian Motion and the Ornstein-Uhlenbeck process. The form of the transition kernels for the CLD are more involved. They can be found in Dockhorn et al. [2021, Appendix B.1].

**Lemma 5.** *The marginal densities $p_t(x)$ of the SDEs treated in Section 2 depend smoothly on $x$ and $t$.*

*Proof.* One can combine the form of $m_t$ and $\Sigma_t$ and the explicit representation of $p_t$ in (11) to see this. More generally, for the Brownian motion and the OU-Process this is a result of the Hörmander theorem. For the CLD it is a result of hypocoercivity. □

## C.2 Form of the drift

We now prove that we can represent the drift as in (12).

**Lemma 6.** *Assume that $p_t$ has the form* (11), *i.e.*

$$p_t(z) = \int_{\mathbb{R}^d} \mathcal{N}(z; m_t(z_0), \Sigma_t)\mu_{data}(z_0)\mathrm{d}z_0.$$

*Then* (12) *holds true, i.e.*

$$\nabla \log p_t = \frac{\nabla p_t(z)}{p_z(z)} = \Sigma_t^{-1}(z - \mathbb{E}[m_t(Z_0)|Z_t = z]).$$

*Proof.*

$$\nabla p_t(z)$$
$$= \frac{1}{\sqrt{\det(2\pi\Sigma_t)}} \int_{\mathbb{R}^d} \nabla \exp\left(-\frac{1}{2}(z - m_t(z_0))\Sigma_t^{-1}(z - m_t(z_0))\right)\mu_{\text{data}}(z_0)\mathrm{d}z_0$$
$$= \frac{1}{\sqrt{\det(2\pi\Sigma_t)}} \int_{\mathbb{R}^d} \Sigma_t^{-1}(z - m_t(z_0))\exp\left(-\frac{1}{2}(z - m_t(z_0))\Sigma_t^{-1}(z - m_t(z_0))\right)\mu_{\text{data}}(z_0)\mathrm{d}z_0$$
$$= \Sigma_t^{-1}z p_t(z)$$
$$- \frac{1}{\sqrt{\det(2\pi\Sigma_t)}} \int_{\mathbb{R}^d} \Sigma_t^{-1}m_t(z_0)\exp\left(-\frac{1}{2}(z - m_t(z_0))\Sigma_t^{-1}(z - m_t(z_0))\right)\mu_{\text{data}}(z_0)\mathrm{d}z_0.$$

If we now divide everything by $p_t$ it cancels in the first summand. In the second summand we get the formula for the conditional expectation (see, for example [Klenke, 2013, Section 8.2]). □

## D Reverse SDEs: The general case

One can also treat more general forward SDEs than we did in Section 1. This leads to a more complicated form of the reverse SDE. Our Theorems do not use the specific form of the forward SDEs and therefore also hold in the general case. We denote the forward SDE by

$$\begin{aligned} \mathrm{d}X_t &= \beta(t, X_t)\mathrm{d}t + \sigma(t, X_t)\mathrm{d}W_t, \\ X_0 &\sim \mu_{\text{data}}, \end{aligned} \tag{18}$$

where $\mu_{\text{data}}$ is supported on $\mathcal{M} \subset \mathbb{R}^d$ and $W_t$ is a $\mathbb{R}^r$ valued Brownian motion. The drift $b$ maps from $\mathbb{R} \times \mathbb{R}^d$ to $\mathbb{R}^d$. The dispersion coefficient $\sigma$ maps from $\mathbb{R} \times \mathbb{R}^d$ to the $d \times r$-matrices. The time-reversed process $Y_t := X_{T-t}$ is then a solution to

$$\begin{aligned} \mathrm{d}Y_t &= b(t, Y_t)\mathrm{d}t + \sigma(T - t, Y_t)\mathrm{d}B_t, \\ Y_0 &\sim q_0, \end{aligned} \tag{19}$$

with

$$b_i(t, y) = -\beta(T - t, y) + \frac{\sum_j \nabla_j(a_{ij}(T - t, y)p_{T-t}(y))}{p_{T-t}(y)}, \quad a(t, y) = \sigma(t, y)\sigma(t, y)^T,$$

and $q_0 = p_T$, see Haussmann and Pardoux [1986]. This simplifies to the case discussed in Section 1 if $r = d$, $\sigma$ is is a multiple of the identity matrix and $\beta$ is independent of the time $t$. In Assumption 1 we treat the case where the SDEs are of the form written-out in Section 1. The items $(i) - (iii)$ need to be replaced by their more general counterpart as found in Haussmann and Pardoux [1986, Section 2]. In the last item $(iv)$, $\nabla \log p_t$ needs to be replaced by $\frac{\sum_j \nabla_j(a_{ij}(T-t,y)p_{T-t}(y))}{p_{T-t}(y)}$.

## E Proofs

### E.1 Proofs of the theorems

We now give proofs of our main results and briefly summarize the key steps in an intuitive way. In our study we would like to include the case when $\mu_{\text{data}}$ is degenerate and supported on a low-dimensional

substructure $\mathcal{M}$. As we have seen in Section **??**, this can lead to an exploding drift in the reverse SDE as $t \to T$. Nevertheless, in order to understand the properties of $\mu_{\text{data}}$, it is crucial to study the properties of solutions to the reverse SDE at time $t = T$. This is where the main mathematical difficulties come from. The proofs are mostly independent of the specific form of the forward SDE and hold for more general forward/backward SDEs than those stated in Section 1, see Appendix D.

### E.1.1 Theorem 1

We now proceed with proving Theorem 1.

*Proof.* Let $P$ be the measure on $\Omega = C([0, T], \mathbb{R}^n)$ induced by the forward SDE (1) started in $p_0 = \mu_{\text{data}}$. $P$ has marginals $p_t$. Denote by $X_t$ the canonical projections $X_t(\omega) = \omega(t)$ for $\omega \in \Omega$. We define $Q$ through

$$\frac{\mathrm{d}Q}{\mathrm{d}P}(\omega) = \frac{\mathrm{d}\mu_{\text{prior}}}{\mathrm{d}\pi_T}(\omega(T)).$$

By the data processing inequality we obtain (see [Liese and Vajda, 2006, Theorem 14]),

$$KL(q_T | \mu_{\text{data}}) \le KL(Q|P) = KL(\mu_{\text{prior}} | \pi_T). \tag{20}$$

It remains to prove that by running $Q$ backwards we obtain a solution to (2) started in $\mu_{\text{prior}}$. We denote the generator of the reverse SDE (2) by $\mathcal{L}$. Denote by $Q^R$ and $P^R$ the time reversals of $Q$ and $P$. Our assumption are such that $P^R$ is a Markov process solving the martingale problem for $\mathcal{L}$ (see [Haussmann and Pardoux, 1986, Theorem 2.1]). A short calculation shows that $Q^R$ is still Markov (see, for example [Léonard, 2011, Proposition 4.2]). Furthermore for $f \in C_c^\infty(\mathbb{R}^n)$,

$$\mathbb{E}_{Q^R}\left[f(X_t) - f(X_s) - \int_s^t \mathcal{L}f(X_r)\mathrm{d}r | X_s\right]$$

$$= \frac{\mathbb{E}_{P^R}\left[\left(f(X_t) - f(X_s) - \int_s^t \mathcal{L}f(X_r)\mathrm{d}r\right)\frac{\mathrm{d}\mu_{\text{prior}}}{\mathrm{d}p_T}(X_0) | X_s\right]}{\mathbb{E}_{P^R}\left[\frac{\mathrm{d}\mu_{\text{prior}}}{\mathrm{d}p_T}(X_0) | X_s\right]}$$

$$= \frac{\mathbb{E}_{P^R}\left[\left(f(X_t) - f(X_s) - \int_s^t \mathcal{L}f(X_r)\mathrm{d}r\right) | X_s\right]\mathbb{E}_{P^R}\left[\frac{\mathrm{d}\mu_{\text{prior}}}{\mathrm{d}p_T}(X_0) | X_s\right]}{\mathbb{E}_{P^R}\left[\frac{\mathrm{d}\mu_{\text{prior}}}{\mathrm{d}p_T}(X_0) | X_s\right]}$$

$$= \mathbb{E}_{P^R}\left[\left(f(X_t) - f(X_s) - \int_s^t \mathcal{L}f(X_r)\mathrm{d}r\right) | X_s\right] = 0.$$

In the second equality we used the Markov property of $P^R$. In the last one we used that $P^R$ solves the martingale problem for $\mathcal{L}$. Therefore also $Q^R$ solves the martingale problem for $\mathcal{L}$. Denote by $Y_t$ a solution to (2) on $[0, T)$. Since solutions to (2) are unique in law on $[0, S]$ for $S < T$ (see [Karatzas and Shreve, 2012, Section 5.2]) and the solutions are continuous, the law of $Y$ is equal to $Q^R$ on $[0, T)$. But the paths of $Q^R$ are continuous on $[0, T]$. Therefore, $Y$ can be extended to $[0, T]$, i.e. the limit $Y_T := \lim_{t \to T} Y_t$ exists almost surely and its distribution is equal to the $T$-time marginal distribution of $Q^R$, which is the 0-time marginal of $Q$. We denote the marginals of $Q$ and $P$ by $Q_t$ and $P_t$. Since $Q$ is absolutely continuous with respect to $P$, $Q_0 = \mu_{\text{sample}}$ is absolutely continuous with respect to $P_0 = \mu_{\text{data}}$. Analogously, if $\mu_{\text{prior}}$ and $p_T$ are equivalent, then so are $P$ and $Q$ and therefore $P_0$ and $Q_0$. This proves $(i)$.

$(ii)$ is a consequence of the data processing inequality for $f$-divergences ([Liese and Vajda, 2006, Theorem 14]), analogous to (20). $\qquad\square$

The main idea of this proof is that we look at the forward SDE for $X_t$ first. It induces a distribution $\mathbb{P}$ over all continuous paths in $\Omega = C([0, T], \mathbb{R}^d)$. If we reverse the time direction of this distribution on $\Omega$, we get a solution to the reverse SDE, started in $p_T$. This reverse solution is well behaved as $t \to T$, since $Y_T = X_0$. The solution for a different initial condition $\mu_{\text{prior}} \neq p_T$ is obtained by reweighting $\mathbb{P}$. This does not change the qualitative behaviour of $Y_T$, which still exists and is well defined. We then use a uniqueness result to see that any solution of (19) inherits these properties.

### E.1.2 Theorem 2

*Proof.* Denote the space $C([0,T], \mathbb{R}^d)$ by $\Omega$. We also define the natural filtration $\mathcal{F}_t = \sigma(x(s)|s \leq t)$. We denote the distribution of $Y$ on $\Omega$ by $\mathbb{P}$. We define $\tilde{\mathbb{P}}$ by reweighting $\mathbb{P}$ with $Z_t$ on $\mathcal{F}_t$. By Girsanov theorem (see [Karatzas and Shreve, 2012, Section 3.5]) we know that the canonical process under $\tilde{\mathbb{P}}$ is a solution to (8) on $[0,T)$. Since $Z_t$ is uniformly integrable, its limit $Z := \lim_{t \to T} Z_t$ exists in $L^1$. Furthermore $\tilde{\mathbb{P}}$ is absolutely continuous with respect to $\mathbb{P}$ on $\mathcal{F} = \sigma(\cup_{t<T} \mathcal{F}_t) = \sigma(x(t)|t < T)$ with density $Z$. We define $x(T) := \lim_{t \to T} x(t)$. Then the event

$$\mathcal{A} = \{x(T) := \lim_{t \to T} x(t) \text{ exists and } x(T) \in \mathcal{M}\}$$

has probability 1 under $\mathbb{P}$ (see Theorem 1) and therefore also under $\tilde{\mathbb{P}}$. Furthermore, $x(T)$ is measurable with respect to $\mathcal{F}$. Therefore the distributions of $x(T)$ under $\mathbb{P}$ and $\tilde{\mathbb{P}}$ are equivalent. The canonical process under $\tilde{\mathbb{P}}$ is therefore a solution of (8), with the property that its time $T$-marginal is well defined and equivalent to the time $T$-marginal of (2). We can use uniqueness in law on $[0,S)$ for any $S < T$ and extend it to $[0,T]$ as in the proof of Theorem 1. This shows that every solution to (8) has the desired properties.

Finally we show that if $e$ is bounded, it fulfils Assumptions 2. We define by $H_t = \int_0^t \|e(s, \hat{Y}_s)\| \mathrm{d}s$. Then there is a Brownian motion $W_t$ such that we can write $Z_t$ as

$$Z_t = \exp\left(W_{H_t} - \frac{1}{2} H_t\right).$$

Since $e$ is bounded by $M$, $H_t$ is bounded by $TM$. In particular, one can view $Z_t = \mathbb{E}[Z_{TM}|\mathcal{F}_{H_t}]$. Therefore $Z_t$ is uniformly integrable since it can be viewed as a family of conditional expectations. $\square$

Here we essentially applied the Girsanov Theorem on $[0,T)$. Using the uniform integrability of the Girsanov weights $Z_t$, we are able to extend it to $[0,T]$. Therefore, we can infer that the distribution of $Y$ and $\hat{Y}$ are actually equivalent on the whole path space $C([0,T], \mathbb{R}^d)$. In particular, their time $T$-marginals will be equivalent too, which is the claim of the theorem.

### E.2 Proof of the Lemmas

We start by proving Lemma 1.

*Proof.* The forward drifts are $\beta(x) = 0$, $\beta(x) = -\frac{\alpha}{2}x$ and $\beta(x,v) = (v, -x - 2v)$ for the Brownian Motion, the OU-Process and the Critically Damped Langevin Dynamics respectively. In particular, these are all linear maps and therefore fulfil conditions $(i)$ and $(ii)$ of Assumption 1.

We show in Appendix C.1 that $\log p_t$ is $C^\infty$ in $t$ and $x$ for $t > 0$. Therefore we can integrate $p_t$ and its derivative over compact sets, implying that condition $(iii)$ holds. Furthermore, the Hessian w.r.t. $(x,t)$ is continuous and obtains its maximum and minimum on the compact set $[S,T] \times B_N$, where $B_N$ is the ball of diameter $N$ around the origin. Therefore the gradient $\nabla \log p_t$ is Lipschitz on $[S,T] \times B_N$, which proves $(iv)$. $\square$

We now prove Lemma 2.

*Proof.* We have that $X_0 \sim \mu_{\text{data}}$. Denote the mean and covariance of $\mu_{\text{data}}$ by $a$ and $C$ respectively. We define

$$n_t = \mathcal{N}(m_t, V_t)$$

for some functions $m_t$ and $V_t$. If $V_t$ would not have full rank, $n_t$ would be a degenerate distribution. Since $p_t > 0$ almost everywhere for $t > 0$, the KL divergence from $p_t$ to $n_t$ would be infinite. We can therefore restrict $V_t$ to be an invertible matrix. We denote the entropy of $p$ by $H$, $H(p) = -\int \log(p)p\mathrm{d}x$.