# OpenReview forum: "Score-Based Generative Models Detect Manifolds"
_NeurIPS.cc/2022/Conference — NeurIPS 2022 Accept_

### Official Review · Reviewer_dRbJ · 2022-07-02

**Rating:** 7
**Confidence:** 2
**Soundness:** 3 good
**Presentation:** 3 good
**Contribution:** 4 excellent

**Summary:**

This is a pure theoretical study of the score based generative model. The authors discover the conditions under which the sample distribution is equivalent to the data distribution.

**Questions:**

Move Theorem 1&2 to later sections, especially after assumption 1&2 are introduced, may make it flow better.

**Limitations:**

No societal impace for a theoretical work.

**Strengths And Weaknesses:**

1. The theorems provided here could help researchers better understand those score based generative models, which is significant to the whole community.
2. The assumptions proposed here can guide future design of score based models.
3. As a pure theoretical work, the authors provide detailed proofs, which is helpful. But the lack of experimental verification could make it hard to digest. Although, I would not penalize the authors for this, since numerical verification on high dimension is a challenge on itself.

---

> ### Author Response · Authors · 2022-08-02
> **Response to Reviewer dRbJ**
>
> Dear Reviewer,
>
> Thank you for the positive review of our work; we are glad that you consider it a good contribution and will help practitioners. We followed your suggestions and moved the Assumptions after the Theorems. Furthermore, we added numerical experiments on CIFAR-10, empirically validating our claims.

---

### Official Review · Reviewer_ifjq · 2022-07-08

**Rating:** 4
**Confidence:** 3
**Soundness:** 4 excellent
**Presentation:** 2 fair
**Contribution:** 2 fair

**Summary:**

The paper considers the approximation errors in the sampling process of score-based models. Specifically, the errors are in two folds: the estimation error of neural networks and the approximation of priors. The authors study the terminal distribution of the reverse SDE under such errors. They first show that the terminal measure is absolutely continuous w.r.t the data measure if we have an imperfect prior. Next, they show that the terminal distribution has the same support as the data distribution, under mild assumptions of the estimation error. Toy experiments demonstrate the validity of the theoretical results.

**Questions:**

I copy and paste the questions above.

Before going into the concrete questions, I would like to double check some terms and the overall proof ideas with the authors. Please correct me if I'm wrong. (1) In line 71 "its distribution is equivalent to the distribution of ..": Does the equivalence here only mean that the two distributions have the same support? (2) The general proof idea of theorem 2: To my understanding, the paper uses the Girsanov theorem with $Z_T$ as the ratio $\frac{\mu_{sample}}{\mu_{data}}=Z_T$. The uniformly integrable martingale assumption is to ensure the boundness of $Z_T$. Together the results manifests itself.



**Limitations:**

The authors have adequately addressed the potential negative societal impact of their work.

**Strengths And Weaknesses:**

## Strengths

- The problem of approximation errors in score models is interesting and the theoretical understanding is lacking. The paper takes a step in this direction and investigates the terminal distribution of reverse sampling with the existence of various sources of errors.

- The paper verifies the correctness of the main results on toy experiments, and examines popular SDEs to see whether they fulfill the assumptions.

**Questions**

Before going into the concrete questions, I would like to double check some terms and the overall proof ideas with the authors. Please correct me if I'm wrong. (1) In line 71 "its distribution is equivalent to the distribution of ..": Does the equivalence here only mean that the two distributions have the same support? (2) The general proof idea of theorem 2: To my understanding, the paper uses the Girsanov theorem with $Z_T$ as the ratio $\frac{\mu_{sample}}{\mu_{data}}=Z_T$. The uniformly integrable martingale assumption is to ensure the boundness of $Z_T$. Together the result manifests itself.

## Weaknesses

- The main result (theorem 2) in the paper is straightforward. It's a direct implication of Girsanov theorem.

- The results seem tangent to the actual score-based generative models. For the image score-based generative models, the terminal $T$during sampling is set to $1e-3\sim1e-5$ due to discontinuity at $T=0$. Critically, the support of $p_{T}(x), T>0$ is $\mathbb{R}^n$, and the results trivially hold in this case. In addition, I ran some quick experiments on CIFAR-10 by adding 1 to the predicted score (the paper also does similar things), and the final results are noisy samples, with very large pixel values.



Due to the limited impact of the theoretical results and the gap to practical models, I think the paper has a lot of room for improvement.

---

> ### Author Response · Authors · 2022-08-02
> **Response to Reviewer ifjq**
>
> Dear Reviewer,
>
> we thank you very much for your comments and suggestions. They helped us to improve the paper and guided us as to which areas need to be addressed more carefully.
>
> ## Questions
>
> - Indeed, equivalence means that the two measures share the same support. We made this clearer in the paper now (line 79). Thanks for telling us that this was unclear in the paper before.
> - We detail more on the full proof idea in the next section of this response. Here we would like to briefly point out a subtle difference between $Z_T$ and $\mu_\text{sample}/\mu_\text{data}$. $Z_T$ is the density ratio on the full path space $C([0,T],\mathbb{R}^n)$. To get the density ratio $\mu_\text{sample}/\mu_\text{data}(x)$ at some point $x \in \mathbb{R}^n$ one would need to integrate out all paths that lead towards $x$.
>
> ## Response to the "Weaknesses" section
>
> We will now first respond to the weaknesses section. The questions will be addressed while doing so. As you correctly say, $p_T$ will have full support for any $T>0$, and thus our results would indeed become trivial in this setting. In other words, the main mathematical difficulties stem from the fact that we assume that $\mu_\text{data}$ is supported on a lower-dimensional subset (we argue in the introduction that this is a realistic assumption for most datasets and, furthermore, the empirical measure is only supported on finitely many points). Precisely because of these observations, however, it seems impossible to deduce generalisation results in the regime when $T>0$— the analysis for $T>0$ is not sharp enough to distinguish different scenarios since p>0 holds regardless of the specifics of the situation. The main mathematical work in the paper is therefore to take the limit $T \rightarrow 0$, enabling the study the generalisation capacities of SGMs. In particular, Girsanov’s theorem as well as existence and uniqueness results for SDEs cannot straightforwardly  be applied and require substantial care and effort (note that the manifold hypothesis leads to an exploding drift $\nabla \log p_t(x)$, see Section 5). We really appreciate the fact that you pointed out the importance of the distinction between $T>0$ and $T=0$ (drastically changing the relevant supports), and changed the introduction accordingly.
>
> In the following, we summarise some of the important technical points when considering the limiting case $T=0$. Our first main result is to show that the reverse SDE is well defined, even when one changes its initial conditions. Furthermore, we see that the reverse SDE will have the same support as $\mu_\text{data}$.
>
> 1. Our second main result employs Girsanov's theorem. Applying the first theorem, we know that the reverse SDE has a solution on $[0,T]$. We then want to apply Girsanov's Theorem to additionally change the drift. This gives us the existence of a solution. Furthermore this solution has the property that it still has the same support as $\mu_\text{data}$. Now we use uniqueness-results to show that any other solution to the SDE also has to have these properties. Since uniqueness results need a well-behaved drift, we can only apply them on $[0,T-\epsilon)$ and not on $[0,T]$ directly, as we would like to. We then use a limiting argument to conclude the proof of Theorem 2.
> 2. (a) Due to numerical issues, the implementations of SGMs are not able to be run or trained up to time $T = 0$ and have to stop slightly before. Nevertheless, these times are normally chosen very small, and the algorithms are an approximation of the SDE on the interval $[0,T]$. Furthermore, some properties are not visible before the finite time. As you say, all the measures will be supported in all of $R^d$, rendering the study of generalisation in the way we did impossible. Lastly, if the SDE will land on a manifold at time $T$, this suggests, that it will be very close to the manifold at time $T-\epsilon$.
>    (b) Thank you for the suggestion to implement the experiments on CIFAR-10. In our revised version, there are analogous experiments to the experiments we did before, but on CIFAR-10. We added 1 to the exact score based on all training examples and got the same results as in the low dimensional toy example. If we understand your experiment correctly, you added 1 to the predicted (trained) score rather than to the theoretical one. Distinguishing between the theoretical and the trained score is crucial: This distinction leads to very different experimental results and in fact is closely related to our key findings, as the the difference between the trained and theoretical score tends to infinity as $T \rightarrow 0$. For further intuition, adding 1 to the trained score leads the SDE into areas in which the were no samples to train the score, and therefore it will not pull the SDE back towards the manifold.
>
>
> Thank you for all the input. We believe it has greatly improved the quality of our paper and hope that you think so too.

---

> > ### Comment · Reviewer_ifjq · 2022-08-09
> > **Straightforward theory, but tangent to the practical situation. The title/story is misleading.**
> >
> > Thank you for the reply and the experiments! However, the theory in the paper is developed for $T=0$ terminal time, which is essentially different from the practically used $T>0$ terminal time. Also, the authors showcase the detection of support by perturbing the (approximated) theoretical scores, which is quite inconsistent with their title "Score-Based Models Detect Manifolds" (trained scores) and the main story for the trained score-based models. Lastly, as the authors admit above ("... it will be very close to the manifold at time $T= \epsilon $.."), the experiments conducted by the reviewer at $T= \epsilon$ should still have clean samples, which is not true.
> >
> > I will elaborate more in the following.
> >
> > The reviewer still cannot see the connection between the main results and the practical score-based models. Practically, for terminal time T>0 that used by score-based models, the support of terminal distribution is Euclidean space, and the integral in Eq.8 is bounded.
> >
> > Yes, the reviewer added 1 to the predicted (trained) score rather than to the theoretical one, and empirically the reviewer observed that Eq.8 is still bounded, but the reverse SDE produces noisy samples. It contradicts the arguments made by the authors "Lastly, if the SDE will land on a manifold at time $T$, this suggests, that it will be very close to the manifold at time $T\to \epsilon$."  or "Nevertheless, these times are normally chosen very small, and the algorithms are an approximation of the SDE on the interval [0,T]".  If so, the SDE with the perturbation to the predicted (trained) score can still "detect" the manifold at time $T=\epsilon \to 0$, which doesn't hold true in practice.
> >
> > The reviewer understands that the difference between the trained and theoretical score tends to infinity when $T\to 0$, but for $T=\epsilon>0$, the integral is bounded. That little step from $\epsilon$ to $0$ changes the dynamics but is infeasible in practice since the score would explode.
> >
> > I keep my score at 4 given the glaring issues.

---

> > > ### Author Response · Authors · 2022-08-09
> > > **Thank you! We want to justify a mildly idealized case theoretically and clarify the scope of our work.**
> > >
> > > Dear Reviewer,
> > >
> > > many thanks for taking the time to submit another comment. We appreciate the raised issues and hope we can clarify them.
> > >
> > > # Justification for studying an idealized case theoretically
> > > Many theoretical papers study some kind of mildly idealized case of an algorithm. By taking limits or neglecting some approximations that need to be done in practice, one is often able to uncover structure that is not visible otherwise.In the end, one of course should connect back to the practical implementations and see if the results still hold if the ideal case is only approximated. This is precisely what we tried to do.
> > >
> > > ## Why we studied the case $T = 0$ for SGMs theoretically
> > > For example, SGM implementations contain many approximations to the theoretical stated version, some of which are: the usage of an discretization scheme, introducing numerical floating point errors at every step, rescaling the images and, finally, running the algorithm until time $T = \epsilon$.
> > >
> > > The structure that we want to see is the support matching properties. As you noted, this structure is not visible at positive $T > 0$. Therefore we study the case $T = 0$. We want to emphasize, that $T = 0$ is not even much of an idealization, but actually how the algorithm is motivated in most works. Only in practice one often cannot actually implement this limiting case.
> > >
> > > In this setting we are able to uncover interesting theoretical properties on how the support of a distribution (the data manifold) is matched by an SGM. Finally, we numerically illustrate that theory correctly predicts the behavior, also in the case where $T = \epsilon$: In Figure 3 and 4, we also only ran the algorithms until time $T = \epsilon = 10^{-3}$ and still get the results that are predicted by the theory.
> > >
> > > # Explanation of the scope of our results
> > > We will now try to make the scope of our results more clear to prevent any misunderstanding on what our paper suggests.
> > > Therefore, we also want to clearly state that pretrained score cannot be used as $\pi_t$ in our paper. The pretrained score $s_\theta$ will normally not be the log gradient of some distributions $\pi_t$. Furthermore, $s_\theta$ will typically untrained in large areas of the space which it has never seen. Nudging it to these areas by adding an error will lead to explosion as observed by the reviewer.
> > > We study the case in which $\pi_t$ the distributions that stem from some forward SDE. In this case, the scores are defined everywhere, and we see derive an explicit condition (Assumption 2) what is needed for support matching (= data manifold matching). Two specifically interesting cases for $\pi_t$ are $p_t$ and $\hat{p}t$, the distributions of the forward SDE when started in the true data generating distribution and when started in the empirical measure respectively. Note that $\hat{p}_t$ is used as training objective for $s\theta$ (see Section 3).
> > > - For $p_t$, we would like Assumption 2 to hold, since the data manifold will be the support of the true data generating function. This is however not testable on real-life datasets, only in the Toy experiments we made. We believe further interesting work however to be, to understand how this Assumption 2 could be approximated for the true data generating distribution.
> > > - For $\hat{p}_t$, the data manifold is equal to the training examples. Here we want the conditions not hold in practice, since otherwise no generalization is possible. In particular, one can derive some simplified conditions, and see that we actually need the score approximation to have an unbounded error to its training objective $\hat{p}_t$, to be able to generalize. In Figure 3 and 4 we showcase that also when the algorithm is approximated numerically and only ran until time $T = \epsilon$, the Theory predicts the behavior accurately.
> > >
> > > # Answering to the individual statements
> > > Finally, we want to answer to some specific remarks in case they are not resolved by the above.
> > > 1. Unfortunately, we do not understand what you mean, when you say that "the experiments conducted by the reviewer at $T = \epsilon$ should still have clean samples, which is not true."  Adding an error to the pretrained score is not in the scope of our results.
> > > 2. "The reviewer still cannot see the connection between the main results and the practical score-based models. Practically, for terminal time T>0 that used by score-based models, the support of terminal distribution is Euclidean space, and the integral in Eq.8 is bounded." We plotted this experiment in Figure 4. The plot goes until time $\epsilon = 10^{-3}$. The blue line explodes towards infinity whereas the orange line stays bounded, as predicted by the theory.
> > >
> > >
> > > We hope that our answer clarifies the applicability of our results. If that is the case we would greatly appreciate if the score could be raised to a non-rejection.

---

> > > > ### Comment · Reviewer_ifjq · 2022-08-09
> > > > **The dynamics of $T>0$ and $T=0$ are completely different**
> > > >
> > > > Thanks for the reply. The reviewer gets stuck with an urgent stuff, and will give more details later on.
> > > >
> > > > I think the idealized $T=0$ case are completely different from $T>0$, both in theory and in practice. I listed many reasons above. The authors attempt to demonstrate that the terminal distribution at $T = 1e-3$ is close to $T = 0$, implying that their theory is still applicable to small $T$.
> > > >
> > > > ### **The argument does not hold.**
> > > >
> > > > I tried to make my point clearer. For $T=1e-3$, adding small perturbation to the predicted score models ($s_\theta$) only produces noisy images empirically. Denote the noisier version as $\hat{s}$. In this case, the difference between $s_\theta$ and the ground-truth score is bounded, and the difference between $\hat{s}$ and $s_\theta$ is certainly bounded. As a result, the difference between $\hat{s}$ and the ground-truth score is bounded. However, we are obtaining noisy samples using $\hat{s}$.
> > > >
> > > >
> > > > ### Minors
> > > >
> > > > - In the authors' response, "Nudging it to these areas by adding an error will lead to explosion as observed by the reviewer." No, it would not explode empirically, but it leads to very noisy samples. Eq.(8) is still bounded.

---

> > > > > ### Author Response · Authors · 2022-08-09
> > > > > **Thanks!**
> > > > >
> > > > > Dear reviewer,
> > > > > we highly value that you took the time to answer again, despite the your urgent matters.
> > > > >
> > > > > ## Notation
> > > > > - Let $t =\nabla \log \hat{p}_t$ be the score of the forward SDE started in the empirical measure on the training examples. (It will be a Gaussian mixture, in case of CIFAR-10 with 50 000 components)
> > > > > - Let $\hat{t}$ be the perturbed true score, $\hat{t} = \nabla \log \hat{p}_t + e$, where $e$ is some constant perturbation, like $e = (1, 1, \ldots, 1)$.
> > > > > - Let $s_\theta$ be the pretrained score
> > > > > - Let $\hat{s}_\theta = s_\theta + e$ be the perturbed trained score.
> > > > >
> > > > > ## What our results predict / What they do not predict
> > > > > 1. For the error between $t$ and $\hat{t}$, Equation 8 is bounded. This can be seen in Figure 4a. (Or simpler, we can deduce that Eq 8 needs to be bounded since the error is bounded everywhere). Therefore we can deduce that Assumption 2 holds and the SGM will not generalize. Indeed, in Figure 3a (orange line) and Figure 3b, we see that also in practice at $T = \epsilon$, the SGM employing $\hat{t}$ does not generalize, as the theory at time $T = 0$ predicts.
> > > > > 2. In case the error between $s_\theta$ and $t$ would be bounded (Or, less restrictive, Eq. 8 would be bounded), $s_\theta$ would only reproduce training examples. Since in Figure 3a we see that $s_\theta$ does generalize (blue line), we can presume that Assumption 2 does not hold. In Figure 4a, we indeed see that also when running the SGM only until time $T = \epsilon$, we see that the Girsanov weights explode and Eq 8 is looks unbounded, as predicted by the Theory at time $T = 0$.
> > > > > 3. We do not say anything about the support matching between $s_\theta$ and $\hat{s}_\theta$. Since $s_\theta$ is just pretrained drift, already possessing an unbounded error to the true empirical score (see point 2), it does not have the properties of a score stemming from a true forward SDE. Therefore our theorems are not applicable. In particular, by adding noise, you will lead the SDE to regions which the drift has not seen yet, leading to noisy samples or samples with very large pixel values as you described in your first comment.
> > > > >
> > > > > We hope we were able to clarify the notation. In case there are specific remarks you would want us to include in the camera-ready version to make the results better understandable, we are very happy to include these.
> > > > > In case we did not understand which Experiment you did conduct and which results you did expect in alignment with our Theory, we are also happy to discuss some more.

---

> > > > > > ### Comment · Reviewer_ifjq · 2022-08-09
> > > > > > **Why unbounded when $T=\epsilon=1e-3$?**
> > > > > >
> > > > > > We are talking about $\hat{s}_\theta$ and $t$ right? The integral in Eq(8) is over the $[1e-3, 1]$ interval. Why does the error between $\hat{s}_\theta$ and $t$ is unbounded?

---

> > > > > > > ### Author Response · Authors · 2022-08-09
> > > > > > > **Thanks for the question**
> > > > > > >
> > > > > > > We are not sure we understood the question right, but will try to answer it. If the question persists, feel free to ask again.
> > > > > > >
> > > > > > > Most of the time we talk about either $t$ and $\hat{t}$ (Point 1 in our last answer) or $s_\theta$ and $t$ (Point 2 in our last answer.
> > > > > > >
> > > > > > > We do not make any real claims about $\hat{s}_\theta$ (see Point 3 in our last answer). Nevertheless, since $\hat{s}_\theta$ is just a bounded perturbation of $s_\theta$, you can expect the N values to looks the similar as for $t$ and $s_\theta$ for small values of $T$. The error will therefore be bounded, and similar to $s_\theta$, $\hat{s}_\theta$ does not memorize the training data and will produce samples outside of the training data set. In this case however these will just be noisy samples, due to the perturbation and our earlier explanations.
> > > > > > >
> > > > > > > The error on $[T = \epsilon , 1]$ won't in fact be infinite. It will just grow so extremely fast, that the explosion for $T \to 0$ is very visible, see Figure 4a. We had in fact to plot everything directly on the log-domain in that figure, so that the numbers still fit in the computers Float64 range.
> > > > > > >
> > > > > > > Were we able to resolve your issues?

---

### Official Review · Reviewer_3Xuq · 2022-07-10

**Rating:** 6
**Confidence:** 3
**Soundness:** 4 excellent
**Presentation:** 2 fair
**Contribution:** 3 good

**Summary:**

This paper is a theoretic paper that solidifies the well-definedness of sample measure and proves the equivalence of the sample measure and the learnt measure under Assumption 2.

**Questions:**

- Is $Q$ in the proof of Theorem 1 well-defined? Does $\frac{\mathrm{d}Q}{\mathrm{d}P}(w)=\frac{\mathrm{d}\mu_{prior}}{\mathrm{d} p_{T}}$ enough to define $Q$? It seems that I am missing some parts, and it would be greatly appreciated if the authors let me know what I am missing.

**Limitations:**

- I am a practitioner in the community of diffusion models. In my perspective, a paper in this venue is better to contain intuitive illustrations and related explanations. In the current form, the submitted manuscript is flawless in its solidness, and I believe this is good work, but I doubt if this version could be valued in this field simply because it is difficult to understand the contents for the audience. Please remember that the expected audience of this paper is mostly the practitioners, and most of them are short-lesson-seekers. For instance, what is the meaning of uniformly integrable martingale? NeurIPS is not Annals of Mathematics, and sufficient interpretation of mathematical concepts is required. With the kind and satisfactory illustrations and insights, practitioners could find the true value of this paper.

**Strengths And Weaknesses:**

$\textbf{Strengths}$
- Until now, there is no guarantee that the reverse diffusion is well-defined at $t=0$ if $p_{data}$ is embedded in a low-dimensional manifold. This paper is the first to prove that we can safely solve the reverse diffusion up to $t=0$ when we use the previously suggested diffusion strategies (VE, VP, CLD).
- On top of that, this paper shows the well-definedness of the generative measure if Assumption 2 holds. Theorem 2 additionally proves that the sample measure and the trained measure have identical support.

$\textbf{Weaknesses}$
- It would have been much novel work if the author proved that Assumption 2 is the necessary and sufficient condition in Theorem 2. Theorem 2 only proves one direction, and it is difficult to conclude that uniform integrability is the key diagnosis of indicating the overfitting of the trained network.

- It would be much better to move the proof of theorems to the appendix and put more explanations and experiments. In particular, this paper significantly lacks the empirical validation of their claims. Although I believe the claims are novel, the one-point lesson is missing to the community of diffusion models. What would be the "concrete truth" observed from the high-dimensional experiment?

$\textbf{Notes}$
- This is not a weakness, but I should note that Theorem 1-(ii) is nothing but Theorem 1 of [**Song21Maximum**]. In Theorem 1 of [**Song21Maximum**], they used the data processing inequality for KL divergence, but the data processing inequality is well known for its $f$-divergence extension, and Theorem 1-(ii) is merely a straightforward generalization of the work of [**Song21Maximum**]. With all this, nevertheless, proving Theorem 1-(i) is a significant improvement, so I value Theorem 1.

- Again, this is not a weak point, but I think it would be preferable to describe the meaning of equivalent measure for the general audience. A measure $\mu$ is equivalent to $\nu$ if $\text{supp}(\mu)=\text{supp}(\nu)$, and this can be framed in an everyday language as "support matching force". The general audience will much more value this paper if Theorem 2 is well-understood.

- Not a weak point, but it seems that Assumption 2 is akin to the assumption of Theorem 1 of [**Bortoli21Diffusion**]. Theorem 1 of [**Bortoli21Diffusion**] explicitly bounds the total variation distance between the data distribution and the generative distribution under the uniformly bounded score estimation. Under such previous research, it would be very interesting to compare Theorem 1 of [**Bortoli21Diffusion**] and the authors' Theorem 2.

- To get a generalizable model, the authors claim that Assumption 2 should be violated. [**Kim22Soft**] introduced Unbounded NCSN and Unbounded DDPM as $\mathbf{s_{\theta}}(x_{t},\eta(t))$ with $\lim_{t\rightarrow 0}\eta(t)=\infty$ instead of NCSN and DDPM that parameterize the score estimation as $\mathbf{s_\theta}(x_t,t)$, to enable the score network to successfully estimate the unbounded score function. But it seems that NCSN and DDPM are more appropriate to violate Assumption 2, from their network design. It could be an interesting topic to investigate the network architectures with respect to the generalization power.

[**Song21Maximum**] Song, Yang, et al. "Maximum likelihood training of score-based diffusion models." Advances in Neural Information Processing Systems 34 (2021): 1415-1428.

[**Bortoli21Diffusion**] De Bortoli, Valentin, et al. "Diffusion Schrödinger bridge with applications to score-based generative modeling." Advances in Neural Information Processing Systems 34 (2021): 17695-17709.

[**Kim22Soft**] Kim, Dongjun, et al. "Soft Truncation: A Universal Training Technique of Score-based Diffusion Model for High Precision Score Estimation." International Conference on Machine Learning (2022).

---

> ### Author Response · Authors · 2022-08-02
> **Response to 3Xuq**
>
> Dear Reviewer,
>
> We thank you for your detailed review and many insights. We improved the paper by implementing some experiments on CIFAR-10 to verify the results empirically. Furthermore, we carefully rewrote the introduction to make clearer what our results imply in practice. We hope that these changes lead to the results being better understandable to a wider audience.
>
>
>
> ## Weaknesses
>
> - Assumption 2 is probably only a sufficient, but not a necessary condition. There could be drifts that are very badly behaved, such that Assumption 2 does not hold, but which still land on the right manifold in the end. The toy examples are all very degenerate (consider a drift that rotates the space infinitely fast) and therefore not very realistic. Nevertheless, our intuition is that there are borderline cases in which equivalence does not hold. If you are interested, we are happy to discuss more during the discussion period.
> - In hindsight we agree that the proofs should be in the appendix and require more explanation, experiments and empirical validation. We have now implemented these changes. The paper structure is also now reordered, see also what we wrote at the beginning of the response.  The main take home message is that even though we minimise a loss between $s_\theta(x, t)$ and $\nabla \log \hat{p}t$ (for a definition of $\nabla \log \hat{p}_t$ see the introduction of the paper), we actually do not want the $s\theta(x, t)$ to be to close to $\nabla \log \hat{p}t$. Furthermore, we identify exactly in which sense $s\theta(x, t)$ and $\nabla \log \hat{p}_t$ should not be close to each other. We hope that this is clearer in the revised version of our paper.
>
> ## Notes
>
> - We included a comparison to Song21Maximum and Bortoli21Diffusion in our introduction (see lines 55-69). The main difference is that both of these papers make much stronger assumptions on $p_0$ and the scores $\nabla \log p_t$ then we do. In particular, both papers assume that $\mu_\text{data} > 0$ everywhere.
> - The work Kim22Soft does seem very exciting, thank you for drawing our attention to it. We do think it is an interesting research direction and understanding how the assumptions relate to the neural network architecture and parameterization are very interesting from a practical and theoretical point. We included this in our outlook at the end of Section 4.2, (lines 212-220).
> - We included the definition of equivalence just before the main result. We forgot this in the first iteration, thank you for drawing our attention to it. With the surrounding discussion, we hope Theorem 2 is better understandable to a greater audience now.
>
> ## Question
>
> - The main point here is that $dQ/dP(w) = d\mu_\text{prior}/dp_T(w_T)$, where $w$ is evaluated only at its final time $T$ (the initial time for the reverse SDE). Basically, we use that the density $dQ/dP$ exists and is well defined. Then all we do is reweight each path by the relative density, which only depends on the final (initial) condition of the path for the forward (reverse) SDE. This gives us a new path measure. We then prove, that the paths from this path measure still fulfil the same SDE (the samples are solutions to the forward/reverse SDE, depending on in which time direction the path is traversed), and actually have the right final/initial distribution ($\mu_\text{prior}$). We hope this clarified the question.
>
> Thank you for all the input. We believe it has greatly improved the quality of our paper and hope that you think so too.

---

> > ### Comment · Reviewer_3Xuq · 2022-08-04
> > **Great work! I want more discussion on this paper.**
> >
> > It seems that the authors revised many parts in their manuscript, so I have read the whole manuscript again and my evaluation will base on the revision no matter how different it is to the original manuscript. Overall, the revised manuscript is very well written for the practitioners, but I raise some more questions to evaluate the paper more precisely. I am so sorry for this. I am fully willing to raise my score if the following questions are properly answered.
> >
> > 1. Is there any theoretic analysis possible to prove the Wasserstein distance between $\mu_{sample}$ and $\mu_{data}$ or $\hat{\mu}_{data}$? Wasserstein distance is weaker than both Total Variation or KL divergences, and it could properly measure the distance between dirac delta measures. I am not trying to burden the authors or require any further theory in this respect. I simply want to discuss if investigating with respect to Wasserstein distance could be properly handled, or not.
> >
> > 2. I should mention that Theorem 1 of De Bortoli is the generalization of Theorem 1-(ii) of this paper. Theorem 1 of De Bortoli proves that the TV distance between the sample distribution (solved via discretization of Eq. 6) and the data distribution is bounded by two terms, where the first term is the distance between the prior and $p_{T}$. The second term is dominated by two factors. The first factor is the discrepancy between Eq. 6 and Eq. 2, and the second factor originates from the discretization error. Therefore, I strongly recommend the authors to mention in the main script that Theorem 1 of De Bortoli is a generalization of Theorem 1-(ii) to some extent. The authors could feel it is somewhat unfair because the point is different, but I think it is nothing to do with the major contribution of this paper, and I think it should be mentioned. The readers who could understand and are interested in this paper would understand the true value of this paper with or without such mention.
> >
> > 3. It seems that Figure 3 is a new figure. In Figure 3-(b), is it in the training dataset? I can't see the point. Is it really true that "All of the plotted images are nearly equal to an CIFAR-10 image"? I only see white-background meaningless images.
> >
> > 4. Section 5 is written based on the Brownian motion. I like this part, and I could easily believe this argument is directly extendable to arbitrary SDEs. Could the authors provide concrete derivations of the analogy of Section 5 with arbitrary SDEs? At least it would much beneficial to include an analysis on VPSDE. Mathematicians sometimes derive some concrete result on a standardized form, and just say the direct generalization of this result to a general form is trivial, and think this is a virtue. But remember, engineers are having hard time deriving the generalization of what mathematicians have made in a standardized form.
> >
> > 5. What is the purpose of Section 6? I highly recommend the authors to focus only on the support and the generalization. This section seems somewhat distract.
> >
> > 6. Is Assumption 1 really needed in the main paper? Combining Lemma 1 and Theorem 1, the result of Theorem 1 holds for any dataset that lies in a compact space. Therefore, I think Assumption 1 is a point that distracts the readers. Even worse, it is not the if and only if condition for the well-definedness (of course I understand how difficult it is to find out the iff condition of that). Therefore, characterizing when the limit exists is a partial contribution for me. Rather, support matching is the main contribution of this paper for me. Thus, well, I recommend postponing Assumption 1 to the Appendix and explaining more about the generalization.
> >
> > 7. Sufficient interpretation of support matching in a benchmark dataset is missing. The only image is illustrated in Figure 3-(b), but this is not enough. I want the realistic CIFAR-10 image to be generated by the perturbed score, as illustrated in Figure 2-(b), but it does not seem to be sampled in the practice, am I right? I believe and understand the strength of this paper, but without the solid generated image, the impact is limited.
> >
> > Minor comment:
> > 1. In line 141 of the revised manuscript, Section number is ??. Don't miss the details!
> > 2. I don't understand what "from appearance to appearance" means in Assumption 1. Could you explain more on this concept?
> > 3. Figure 2 has no caption.
> > 4. DDPMPP -> DDPM++
> > 5. Is lines 245 - 250 really needed?

---

> > > ### Author Response · Authors · 2022-08-08
> > > **Thank you for the feedback!**
> > >
> > > Dear Reviewer,
> > >
> > > thank you for the detailed and helpful feedback! We are glad that you appreciate the changes we already implemented and are happy to discuss more.
> > >
> > > 1. Yes, indeed we also think that a theoretical analysis of the Wasserstein-distance would be a very interesting result. However we believe that this may not be straightforward under the manifold hypothesis and would be best investigated in detail in future work.
> > >
> > > 2. The main difference to De Bortoli's paper is that they do not assume the Manifold hypothesis but assume that $p(x) > 0$ everywhere, which is different to our setting. We compare our paper to De Bortoli in the introduction and have now added a sentence specifically for Theorem 2, see line 57. We hope this makes the comparison more clear.
> > >
> > > 3. We modified Figure 3b to make it clearer. The datapoints are sampled with the true empirical drift (we can evaluate $\hat{p}_t$ as the mixture of 50 000 Gaussians, therefore we can also evaluate the gradient log), but we added an error to the drift. As also seen in Figure 2b, the error "nudges" $\mu_\text{sample}$ towards specific areas, but does not change the support (which in this case are training examples). The same can be seen for the CIFAR-10 images. The samples are still from the CIFAR-10 dataset, but do not follow a uniform distribution anymore. We updated Figure 3b, its caption, and also the caption of Figure 2b and hope the message is clearer now. We also plotted the CIFAR-10 images as comparison to clarify that these images really are "nearly equal to a CIFAR-10 image".
> > >
> > > 4. We are happy that you are interested in this argument and Section 5 now directly treats the general case. This comes at the cost of making the section a bit more technical, but we hope that it is still understandable.
> > >
> > > 5. The errors either stem from the error in the drift or the error in the initial conditions. Section 6 is mainly to show that we can expect the error in the initial conditions to be controllable. Especially, for the OU-Process (VPSDE) and the CLD, we know that the distance towards the normal distribution decreases at an exponential rate. For the Brownian Motion, no such results are known, and it is not clear why the proposed prior distributions work. We derive a new result showing the speed of convergence of a Brownian motion to a normal distribution with the optimal mean and covariance. This justifies the choice of this normal distribution as prior distribution in SGMs employing a Brownian motion (VESDE). Nevertheless, we shortened this section considerably to make space for a longer Section 5 and extended Figure captions for Figure 2b and 3b.
> > >
> > > 6.  Theorem 1 and Theorem 2 both hold for general SDEs, as long as they fulfill Assumption 1 and 2. If we move Assumption 1 to the Appendix, we would need to use "for all SDEs in Section 2", instead of "for all SDEs fulfilling Assumption 1" in the Theorems. While we see that Assumption 1 is technical, we also want to make it clear that our results hold in general, and that the SDEs we discuss are just some common examples. We articulate this point even more since currently there are many suggestions for alternative SDEs to use and our results probably also hold for a broad range of SDEs. We hope that you can share this judgement.
> > >
> > > 7. We hope that this was clarified by point 3. To summarize, our reasoning is as follows:
> > > - In Figure 3a we see that samples that are generated with $\nabla \log \hat{p}_t$ all have a distance of $0$ to the CIFAR-10 training examples (see the orange line), therefore we will only obtain training examples if we sample using the true empirical score $\nabla \log \hat{p}_t$. If we use the pretrained score however, the average distance to CIFAR-10 stays considerably above 0 and therefore generalization happens.
> > > - Figure 3b: See point 3.
> > > - Figure 4a: We see that Novikovs condition holds for the score with constant perturbation. Therefore, the theory (Theorem 2) predicts that the perturbed score will not generalize and only generate more training examples, in line with the empirical results from Figure 3.
> > > - Figure 4b: As comparison we only plot the difference between $\nabla \log \hat{p}_t$ and the perturbed score (orange line) and $\nabla \log \hat{p}_t$ and the pretrained score (blue line). We see that for most of the time interval, the orange line is actually above the blue line, depicting a higher error. Still, there is no generalization happening, since the Girsanov weights do not explode in this case.
> > > 8. We implemented all the minor comments. We excluded the "from appearance to appearance". It means that the $C$ in each line can take a different value (change from appearance to appearance), but we see that this might be confusing.
> > >
> > > Thank you for reading this quite lengthy response. We again feel that our paper has significantly improved from the comments and suggestions. In particular we hope that the changes in Figure 2 and 3 help to clarify the main messages.

---

> > > > ### Comment · Reviewer_3Xuq · 2022-08-09
> > > > **Leaning towards accept**
> > > >
> > > > I believe the reviewer's response resolves most of my concerns. I raise the score from 5 to 6, because 6 represents a paper whose impact is moderate-to-high. At its current status, this paper is lack an outstanding contribution because solid empirical evidence is missing. I look forward to the authors analyzing more of the experiments to support their claims in the camera-ready version. For instance, I want more understanding of the generalization in diffusion models.

---

### Official Review · Reviewer_YBBA · 2022-07-16

**Rating:** 6
**Confidence:** 2
**Soundness:** 3 good
**Presentation:** 1 poor
**Contribution:** 3 good

**Summary:**

This paper is a theoretic investigation on how to learn a manifold with a score-based generative model (SGM). SGM uses an approximated score of the distribution in the middle of the reverse diffusion. This score estimation is only an approximation by a neural network, and this paper provides the study on error bound for this approximation. Also, this paper analyzes the impact of the limited sample population in learning the score function. Another story line is the limitation of diffusion in T because the diffusion step needs to be infinite to make the diffused distribution be a standard Normal distribution. This becomes infeasible in actual implementation, so there should be discrepancy between the final timestep distribution and the prior distribution. This discrepancy is analyzed, and its bound is suggested.

**Questions:**

See the weakness section

**Limitations:**

I think that this paper would need further toy example experiments and plain explanations to secure audience who may use diffusion models without enough knowledge on authors' discussions background.

**Strengths And Weaknesses:**

Strength:

1.
This paper provides a theoretic discussion on many mathematical assumptions suggested by SGMs.

2.
This paper clearly shows the necessary and sufficient assumptions to make the SGM identify the low-dimensional manifold.

Weakness:

1.
This paper is very difficult to understand because of its eccentric structure. I partly understand the authors' effort because this paper is dedicated to the theoretic analyses. However, this paper is really needs a restructuring to draw more attention from possible audience.

1) Assumptions are referred before their appearances.
2) The motivation of Theorem 1 and 2 need to be further provided by formally throwing a research question.
3) Need a further clear statement on error bounds. Assumption 2 shows the characteristics of e(x,t), but I expected the explicit error bound for e.

2.
The score approximation error will grow as we bring the diffused distribution to the standard Normal distribution, i.e. t->T. This is briefly mentioned in line 206-211, and I think that authors might produce further discussions and plain explanation on the interaction of score approximation quality and closeness between the prior and the diffused distributions.

---

> ### Author Response · Authors · 2022-08-02
> **Response to YBBA**
>
> # Response
>
> Dear Reviewer,
>
> thank you very much for your review and feedback.
>
> ## Weaknesses
>
> - We rewrote the introduction to make it more clear how our results can be interpreted. Furthermore, we followed your advice and reordered most of the sections to make the message more clear. We discuss the problems and questions we want to solve in the introduction and end the introduction with an overview of how we proceed. The theorems are only stated in Section 4, after the assumptions are stated and discussed.
> - The results work by showing how the error has to look for the data manifold to be learned or the training examples to be memorised. A specific example in which the Assumptions hold would be of a bounded error. Other than that there are other ways to check if the Assumptions hold, which we also make more clear in Section 4.
> - In this current work we have shown that if the score approximation has specific properties (mainly Assumption 2 is fulfilled), then the sample distribution will learn the data manifold or memorise the training data (depending on which case you apply the theorems to). This can be seen as the first step on the way to prove more quantitative statements that can directly relate the approximation quality in the drift to a distance bounds, even when the initial distribution is supported on a manifold.
>
>  Thank you for your feedback. We believe that the reordering of the paper and the greater focus on experiments and explanations instead of proofs has made the paper a much more valuable addition to the research community and hope you think so too.

---

### Meta-Review · Area_Chair_Uxbs · 2022-08-29

**Recommendation:** Accept
**Confidence:** Certain

**Metareview:**

This paper presents a theoretical analysis of score-based generative models (SGMs) [diffusion models]. Specifically, the paper theoretically studies the effect of approximations used by SGM [1. approximating p_T by µ_{prior} and 2. approximating ∇log(p_t) by a neural network], which currently lacks a solid understanding. The paper presents conditions that assures SGMs can sample from the underlying data manifold and also analyzes conditions under which an SGM memorizes the training data (the latter relates to understanding the generalization properties of SGMs).
Besides technical discussions and clarifications during the rebuttal period, the authors overhauled the introduction section and also added some experiments with CIFAR-10 dataset to support their theory, both of which were requested by the reviewers to enhance the paper. Reviewers were satisfied with the responses and the improvements in the revision. In concordance with them, I believe the paper provides a solid theoretical contribution to our understanding of SGMs and recommend accept.


**Award:**

No

---

### Decision · Program_Chairs · 2022-09-14

Accept